evolution

plumage, hummingbirds, sex-limited polymorphism, social signalling, social selection

**Author for correspondence:**
Eleanor S. Diamant
e-mail: eldiamant@ucla.edu

# Male-like female morphs in hummingbirds: the evolution of a widespread sex-limited plumage polymorphism

Eleanor S. Diamant[1,2], Jay J. Falk[3] and Dustin R. Rubenstein[1]

[1]Department of Ecology Evolution and Environmental Biology, Columbia University, 1200 Amsterdam Avenue, New York, NY 10027, USA
[2]Department of Ecology and Evolutionary Biology, University of California Los Angeles, CA 90095, USA
[3]Department of Neurobiology and Behavior, Cornell University, 215 Tower Road, Ithaca, NY 14853, USA

ESD, 0000-0003-1727-5855; DRR, 0000-0002-4999-3723

Differences in the way males and females look or behave are common in animals. However, discrete variation within sexes (sex-limited polymorphism) also occurs in several vertebrate and invertebrate lineages. In birds, female-limited polymorphism (FLP) in which some females resemble males in coloration is most prominent in hummingbirds, a group known for its morphological and behavioural sexual dimorphism. Yet, it remains unclear whether this intrasexual colour variation in hummingbirds arises through direct selection on females, or indirectly as a non-adaptive byproduct resulting from selection on males. Here, we analysed specimens from more than 300 hummingbird species to determine the extent, evolutionary history and function of FLP. We found that FLP evolved independently in every major clade and occurs in nearly 25% of hummingbird species. Using phylogenetically informed analyses, we rejected non-adaptive hypotheses that FLP is the result of indirect selection or pleiotropy across species. Instead, FLP is associated with ecology, migratory status, and marginally with social dominance, suggesting a socioecological benefit to females. Ultimately, we show that FLP is not only widespread in hummingbirds and likely adaptive, but may also be useful for understanding the evolution of female ornamentation in systems under strong sexual selection.

## 1. Background

Sexual dimorphism in secondary sexual characteristics is typically associated with sexual selection [1,2], particularly in polygynous species where males do not contribute parental care [3–5]. In many animal taxa, there is strong selection on females to look cryptic to avoid predation and on males to be ornamented for mate attraction [6]. However, female ornamentation can occur as a result of female–female competition for resources [7] in systems with high female reproductive skew [8] and theoretically even when sexual selection is driven by females [9], suggesting that social selection, rather than male mate choice, has played a prominent role in the evolution of female ornamentation [10]. Variation within the sexes can also exist in the form of sex-limited polymorphism (i.e. polymorphism existing in only one of the sexes), complicating our understanding of the processes that generate phenotypic variation and patterns of sexual dimorphism. Although sex-limited polymorphism is most common in males, it can also occur in females [11]. Such female-limited polymorphism (hereafter FLP) in which females of a given species express either male-like coloration (androchrome morph) or a distinct alternative (heterochrome morph) occurs in a variety of dimorphic taxa, including insects, fishes, lizards and birds [12–17]. Understanding why FLP exists in species where selection for males and females

to look different is typically strong may provide insights into the evolution of female ornamentation and sexual dimorphism more generally by determining why ornamentation might be selected over crypsis in females.

A variety of adaptive and non-adaptive hypotheses have been proposed to explain the maintenance of FLP. Adaptive explanations for FLP are similar to those for the evolution of ornaments and weaponry in females [8,18–21]. That is, the male-like or female androchrome morph may be the result of *sexual conflict* due to male harassment in which frequency-dependent selection decreases male harassment on the less common morph [12,13,22,23], *sexual selection* due to male mate choice [24] (but see [13]), or *social selection* (i.e. intra- and intersexual competition over resources) to mediate interactions in which androchromes benefit from increased social success or decreased conflict through social signalling [14,25]. However, FLP could also be a non-adaptive byproduct of *sexually antagonistic intralocus conflict* where females resemble males due to strong selection on males and a shared genetic architecture between the sexes [26,27]. Here, FLP would be an intermediate state and resolve as sexual dimorphism once ornaments become sex-linked [27]. Additionally, FLP could simply result from *pleiotropy* whereby selection on other andromorphic traits leads to expression of non-adaptive androchromic variation in females [12].

Explaining the evolution and maintenance of FLP requires a taxonomic group in which the trait is widespread. Among the nearly 10 000 species of birds, FLP with discrete morphs has been mostly documented in hummingbirds (family Trochilidae) [16,28,29]. Interestingly, hummingbirds are often model systems for studying sexual selection because only females care for nests and young, suggesting strong sexual selection on males [30,31]. Of the more than 300 species of hummingbirds [32], only 43 species have been examined for evidence of FLP, with just 19 species (44%) exhibiting the trait [16,29]. Although it remains unclear why FLP exists in such a high proportion of hummingbirds species, in *Heliangelus exortis exortis*, a negative association between androchromic plumage and bill length is hypothesized to underly a more territorial and aggressive foraging strategy by androchromes [33]. However, existing data on sexually dimorphic hummingbird behaviour are consistent with both the pleiotropy (non-adaptive) and social selection (adaptive) hypotheses. That is, since species with sexual dimorphic resource use and bill size often exhibit behavioural dimorphism in which males are more territorial [30,34], they may use androchromic plumage as a social signal to mediate territorial conflict. However, the adaptive social selection hypothesis has not been tested and the association of androchromic plumage and andromorphic bill length may be due to non-adaptive pleiotropy rather than adaptive evolution. By more deeply understanding the extent and evolutionary history of this trait, predictions of non-adaptive hypotheses can be tested more directly rather than assuming an adaptive explanation.

Here, we establish the pervasiveness and evolutionary origin of FLP in hummingbirds using museum specimens from nearly every species in the family, and then explore adaptive and non-adaptive hypotheses underlying this trait in a phylogenetically informed context. We begin by quantifying the prevalence of FLP within hummingbirds and performing ancestral state reconstruction to determine how many times the trait has evolved within the group. Next, we test the two non-adaptive hypotheses for FLP. First, since sexually antagonistic intralocus conflict should resolve through the eventual evolution of complete sexual dichromatism [27], if FLP is a non-adaptive byproduct

of *sexually antagonistic intralocus conflict*, we predict higher transition rates from sexual monochromatism to FLP and from FLP to sexual dichromatism than from sexual dichromatism to FLP or from FLP to sexual monochromatism. Second, we test the *pleiotropy* hypothesis by examining the relationship between the degree of androchromic plumage and bill length, a well-studied trait known to differ between sexes across species, underlying resource use and a sex's ecological niche (i.e. [35–37]). Although wing length and body size are generally thought to relate to resource use in hummingbirds [38,39], these traits are secondary to bill shape and size, which more directly reflect differences in ecological niche [35–37]. If FLP results from pleiotropy, we expect an association between androchromic plumage and andromorphic bills within each species. Finally, to broadly test the *adaptive socioecological hypotheses* underlying FLP in hummingbirds, we use phylogenetic comparative analysis to identify trait associations within and across species that may be either ecologically (e.g. mean temperature, mean precipitation, temperature predictability, and precipitation predictability aggregated over a species' range) or socially (social dominance, body length, migratory status) relevant. These socioecological traits relate to potential inter- and intraspecific social competition, and thus can act as a proxy in exploratory analyses on the relationship between social competition and FLP. Ultimately, this study provides the most species-rich comparative study to date of FLP in any taxonomic group, illuminating the pervasiveness and potential complex function of variable female ornaments in a family of birds that has become a model for the study of ornamentation and sexual selection.

## 2. Methods

### (a) Specimens

We assessed Trochilidae specimens ($N = 16 542$; electronic supplementary material, figure S1) across all recognized species for which there are skins ($N = 307$) at The American Museum of Natural History, the Louisiana State University Museum of Natural Science, National Museum of Natural History, The Field Museum of Natural History, and The Natural History Museum of Los Angeles.

### (b) Sexing

We evaluated specimens clearly labelled as 'male' or 'female' on the tag, including those in which sexing methods were unknown (60% of samples) or for which gonad data were present on the tag (38% of samples). We did not evaluate specimens that were labelled as 'sexed by plumage' (2% of samples). Analyses comparing specimens without gonad data to those sexed reliably suggested that individuals sexed with unknown methods were done so reliably (see electronic supplementary material, text S1). Nonetheless, we took the uncertainty of sexing reliability into account in our analyses by using thresholds as a form of sensitivity analysis when classifying species as exhibiting or not exhibiting FLP (see below for details).

### (c) Quantifying female-limited polymorphism in hummingbirds

All assessed specimens were qualitatively scored on an ordinal scale of androchromic plumage by one researcher (E.S.D.) following Bleiweiss [16,40]. Each specimen—independent of sex—was assigned to a plumage class based on the percentage of sexually dimorphic plumage patch size and colour that were androchromic: Class 1 was defined as 0% androchromic; Class 2 was

defined as less than 25% androchromic; Class 3 was defined as less than 50% androchromic; and Class 4 was defined as greater than or equal to 50% androchromic. To generate an ordinal and categorical classification, Classes 1 and 2 were categorized as heterochromes (i.e. less than 25% of sexually dimorphic plumage traits were androchromic), and Classes 3 and 4 as androchromes (i.e. greater than or equal to 25% of sexually dimorphic plumage traits were androchromic). Ordinal classifications helped account for potential misinterpretation based on human observation and specimen quality. Expected colour types for each sex were determined for each species using field guides [30]. Since plumage patches that are sexually dimorphic differ across species, they were determined on a species-by-species basis. In instances where sexually dimorphic plumage differed by subspecies, plumage descriptions of the nominate subspecies were used for comparison if other subspecies exhibited fewer colour patches or feathers that differed between the sexes. When subspecies differed in colour rather than brightness, females were compared to males of their own subspecies. If males and females differed in more than one decipherable plumage trait, each had equal weight in determining how androchromic each bird was (see electronic supplementary material, text S2 for examples). Unimodality in the distributions of scores for each species was tested (electronic supplementary material, text S3) to differentiate between variation around a heterochromic form from true FLP that should exhibit distinct morphs, and thus, bimodal variation.

Additional specimens from species already present in the dataset ($N = 532$ individuals from $N = 39$ species) were examined at the Moore Lab of Zoology by a second investigator. These specimens were only classified as 'heterochromes' (Class 1 or 2) or 'androchromes' (Class 3 or 4) and not used in frequency histograms or unimodality tests; they were only used to classify FLP for phylogenetic analysis.

Here, we classified a species as *exhibiting* FLP when greater than or equal to 10% of females were androchromic and were not distributed unimodally. We classified a species as *lacking* FLP if greater than 18 female specimens have been sampled (an 85% chance of sampling an androchrome female morph if greater than or equal to 10% of females of a given species exhibit this trait in the population at large), and if that species has not been classified previously as exhibiting FLP. Those species that were categorized as lacking FLP were further classified as being either sexually dichromic or sexually monochromic based upon qualitative assessments of specimens and field guides [30]. Phylogenetic comparative analyses were conducted on eight data matrices with varying presence and absence thresholds as sensitivity analyses (electronic supplementary material, text S4) to determine if more conservative or more liberal thresholds impacted results.

We chose to present a more conservative analysis of FLP presence in the text because it better takes into account risks associated with mis-sexing. Additionally, we chose a moderate absence threshold to balance minimizing excessive branch pruning while also minimizing the risk of falsely classifying a species as lacking FLP. Although this absence threshold could have incorrectly included more species that were mislabelled as lacking FLP, the chances of this are relatively low given that the majority of hummingbird species did not exhibit FLP. Such a threshold also decreased excessive pruning of the tree, which is known to bias transition rates and node states [41]. Importantly, the results from sensitivity analyses are qualitatively similar (see electronic supplementary material, figures S5–S13).

## (d) Evolutionary transition rates: a test of sexually antagonistic intralocus conflict

To determine the number of independent evolutionary origins of FLP in hummingbirds and whether or not FLP is an intermediate plumage stage between sexual monochromatism and dichromatism, we reconstructed ancestral character states and transition rates. Three discrete trait classes (FLP, dichromic and monochromic) were mapped onto a previously published molecular phylogeny [42], which was chosen due to its sampling richness in species (284 species) and loci (four nuclear and two mitochondrial genes) relative to other studies [43,44].

We estimated the root state without predicting the root state likelihoods. Since we did not fix a root state, we instead conducted an ancestral state reconstruction and estimation using a continuous-time Markov chain model to determine the root state likelihoods using the R [45] packages *ape* [46], *geiger* [47] and *corHMM* [48]. We used an equal rates model (ER) (i.e. all transition rates between states are equal), a symmetrical model (SYM) (i.e. transition rates are equal forward and backward from a given state), and an all rates different model (ARD) (i.e. transition rates between states are different). These models were evaluated for fit using Akaike information criterion with correction for small sample sizes ($AIC_c$). For each dataset, we used the transition rates model that had the lowest $AIC_c$ score and considered all where $\Delta AIC_c$ is less than 2. We then used the results found in *corHMM* to reconstruct trait evolution on trees in *ape* and estimated the likelihoods of root states. Three plumage states were estimated at each node in the model: sexual monochromatism, sexual dichromatism, and FLP. Eight ancestral state reconstructions were conducted using the eight matrices that varied in their thresholds for identifying FLP. All species that were not present in our datasets were pruned from a given tree, resulting in final trees ranging from 152 to 198 species, depending on which dataset was used (electronic supplementary material, table S2). We then used the most-supported transition rate model to test for non-adaptive sexually antagonistic intralocus conflict. We estimated the standard deviation of the likelihood of root states by randomly pruning 10% of species over 1000 simulations to determine if tree topology strongly affected ancestral state reconstruction.

## (e) Morphological correlations of androchromic variation: a test of pleiotropy

To test the non-adaptive pleiotropy hypothesis, we analysed photographs of male and female specimens that ranged from heterochrome to androchrome in 22 species (including 11 species with FLP; all were sexually dichromic) across all hummingbird clades. We also measured the wing and bill lengths (see below for details) of each specimen. We constructed a light box and photographed individual specimens with standardized settings from five angles: dorsal, ventral, each side, and with the bill perpendicular to the camera lens. A few species had additional photographs taken if iridescence was not visible from these angles. A white balance card was placed in the field of view and used for white balance normalization in RawTherapee [49]. On average, we assessed 10.3 androchromic males, 6.2 Class 1 heterochromic females, and 11.2 females that vary across Classes 2–4 per species; sample size varied based on specimen availability at The American Museum of Natural History.

To quantify the degree of androchromic coloration and bill length, each species' specimen's photographs were analysed in IMAGEJ [50] by a single investigator. We extracted the mean red, blue and green colour values of 17 plumage patches across the body and measured bill length (see electronic supplementary material, text S5 for detailed methods). We did not measure UV reflectance, which may have limited our ability to detect chromatic differences perceived by birds [51]. However, UV has been found to be qualitatively similar between androchrome females and males in four hummingbird species [52], and in one species exhibiting FLP where UV reflectance was quantified (*Florisuga mellivora*), UV was highly correlated (greater than 97%)

with visible blue in plumages with high UV reflectance (J. Falk 2019, unpublished data).

Next, we trained a linear discriminant analysis (LDA) within each species using the R package *MASS* [53] to extract linear discriminant (LD) values that quantified continuous variation across heterochromic to androchromic females (electronic supplementary material, text S5). This method was successfully able to differentiate androchrome males from Class 1 females for 16 species, which we subsequently used in analyses. This model failed for other species ($N = 6$), which were then excluded from subsequent analyses. We fit a linear model of relative bill length (bill length minus mean bill length of the species) with LD colour values to determine if androchromic plumage is associated with andromorphic bills and the slope of this relationship for each species. Accounting for multiple tests, we established an $\alpha = 0.0031$. Finally, we calculated Blomberg K [54] to determine if there was a phylogenetic signal in the slope of the relationship between andromorphic bill length and androchromic plumage. We conducted similar analyses using wing length and bill length divided by wing length (electronic supplementary material, text S6).

### (e) Trait associations of female-limited polymorphism: a comparative test of adaptive function

We chose three socially relevant traits for which species-level data were available: migratory status, social dominance, and body size ($N = 167$ species). Each species was categorized as migratory or not based on classifications from [55]. Social dominance and body size (quantified as length from bill to tail tips from 6 to 22 cm) data were adopted from [56], which defined behavioural dominance as species that have been described in the literature as 'most aggressive' or 'notably aggressive' towards heterospecific hummingbirds in their community or were able to 'monopolize resources'. In a second model we considered four ecological variables related to climate (mean temperature, mean precipitation, temperature predictability, and precipitation predictability [57,58]) by calculating the aggregate mean of each variable across each species' range [55] ($N = 290$ species) (electronic supplementary material, text S7). Using *caper* [59], we ran linear models with phylogenetic contrasts. Our dependent variable was the binary variable of 'FLP' or 'no FLP' given the classifications from the analysis presented in text with conservative presence and moderate absence thresholds. Our independent variables were migratory status (binary), social dominance (binary) and body size (continuous). Migration can alter hummingbird social structures; some—though not all [60]—species have been noted to separate by sex during migration [61,62], thereby decreasing intersexual competition outside of the breeding season [63]. Although our initial model included monochromic, dichromic and FLP species (with the binary 'no FLP' variable including monochromic and dichromic species), we also ran a model excluding sexually monochromic species for both our analysis of socially relevant traits and ecologically relevant traits (electronic supplementary material, text S7). We excluded sexually monochromic species for two reasons. First, we could better test the binary difference between sexually dichromic and FLP species. We expected that there may be differences in the behavioural traits of sexually monochromic and sexually dichromic species such that potential differences in comparison to FLP species would be obscured. Second, monochromic drab and monochromic ornamented species may be under fundamentally different types of social and ecological selection, and we were concerned that including monochromic species as a whole would confound potential results. By limiting our analysis to sexually dichromic species, we could better test the associations with the evolution of female ornaments among FLP species in comparison to non-FLP.

## 3. Results

### (a) Quantifying female-limited polymorphism in hummingbirds

We found that 209 of 307 hummingbird species met either the presence or the absence threshold. Of these, 47 (41% of non-monochromic species; 23% including monochromic species) exhibited FLP. The two species not included in the ancestral state reconstruction and that exhibited FLP were not present in the phylogenetic tree [42]. FLP was present in every major hummingbird clade (figure 1; summarized in table 1). Although ancestral state reconstruction has an inherent degree of uncertainty, the most-supported ancestral state reconstruction showed that FLP evolved independently roughly 28 times (nodes with greater than 50% likelihood as 'FLP'), as well as in each major hummingbird clade (summarized in table 1; figure 1; see electronic supplementary material, figure S4 for cladogram and figures S5–S13 for sensitivity analyses). The root state was $0.50 \pm 0.028$ s.d. likelihood sexually dichromatic, $0.30 \pm 0.029$ s.d. likelihood sexually monochromatic, and $0.20 \pm 0.023$ s.d. likelihood FLP. The low standard deviations suggest that tree topology did not strongly affect ancestral state reconstruction.

### (b) Evolutionary transition rates: a test of sexually antagonistic intralocus conflict

Estimating transition rates allowed us to determine whether FLP is an intermediate state between sexual monochromatism and dichromatism, as predicted by sexually antagonistic intralocus conflict and the evolution of sexually dimorphic male ornaments (electronic supplementary material, figure S14a) [26,27]. The $AIC_c$ score for the ER model ($AIC_c = 360.47$) was lower than that of the all rates different matrix ($AIC_c = 367.15$) and the symmetrical rates matrix ($AIC_c = 362.95$), indicating that transition rates between plumage dichromatism, monochromatism, and FLP were similar ($0.90 \pm 0.18$ s.e.; electronic supplementary material, table S1; figure S14d). Furthermore, none of our sensitivity analyses resulted in an ARD with higher transitions from sexual monochromatism to sexual dichromatism and from sexual dichromatism to FLP (electronic supplementary material, table S1 and figure S14). Some of our sensitivity analyses resulted in the symmetric rates matrix being more supported when there were higher transition rates from monochromic to FLP and from FLP to dichromic than from monochromic directly to FLP. However, these models demonstrate higher transition rates from FLP to dichromic and dichromic to FLP than other transitions (electronic supplementary material, figure S14g and S14i–k). The only sensitivity analysis in which the all rates different matrix was supported found the highest transition rates from dichromic to FLP (electronic supplementary material, table S1 and figure S14h); though, for this analysis, $AIC_c$ scores did not distinguish between the all rates different, the symmetrical rates, and the equal rates matrices (electronic supplementary material, figure S14f–h). Together, these results do not support the non-adaptive hypothesis that FLP is an intermediate state and that dichromatism is a resolved state of intralocus conflict.

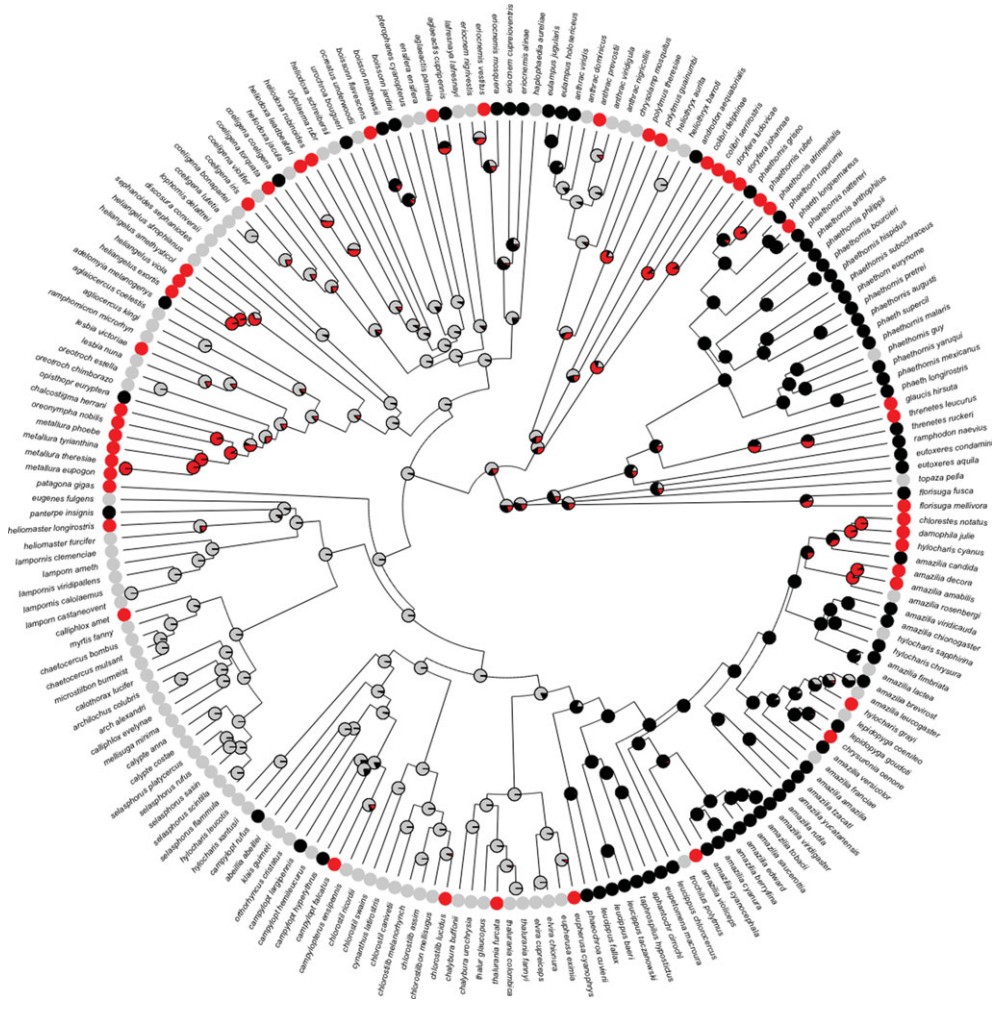

**Figure 1.** Ancestral state reconstruction for plumage state using the analysis where 'presence of FLP' was determined when greater than or equal to 10% females are androchromes and 'absence' when greater than 18 females sampled (greater than or equal to 85% chance of sampling at least one androchrome if 10% of females are androchromes). Pie charts at nodes present the likelihood of being in each plumage state: sexually monochromic (black), sexually dichromic (light grey) and female-limited polymorphism (red). Circles at tips indicate current plumage states for each species. (Online version in colour.)

**Table 1.** Classification of species that exhibited female-limited polymorphism (FLP), did not exhibit FLP, and were not able to be classified with our 'presence' and 'absence' criteria as well as number of independent instances that FLP evolved in each clade. 'Presence of FLP' was determined when greater than or equal to 10% females are androchromes and 'absence' when greater than 18 females sampled (greater than or equal to 85% chance of sampling at least one androchrome if 10% of females are androchromes).

| clade | number of species tested | number (%) of species exhibiting FLP | number (%) of species not exhibiting FLP | number (%) of species not classified | number of independent evolutionary events of FLP |
|---|---|---|---|---|---|
| Topaz | 4 | 1 (25%) | 2 (50%) | 1 (25%) | 1 |
| Mango | 25 | 7 (28%) | 10 (40%) | 6 (24%) | 3 |
| Hermit | 33 | 5 (15%) | 25 (76%) | 3 (9%) | 3 |
| Brilliant | 41 | 7 (17%) | 21 (51%) | 13 (32%) | 6 |
| Coquette | 54 | 11 (20%) | 16 (30%) | 27 (50%) | 3 |
| *Patagona* | 1 | 1 (100%) | 0 (0%) | 0 (0%) | 1 |
| Mtn Gem | 16 | 2 (13%) | 10 (63%) | 4 (25%) | 1 |
| Bee | 37 | 1 (3%) | 16 (43%) | 20 (54%) | 1 |
| Emerald | 96 | 13 (14%) | 62 (65%) | 21 (22%) | 9 |

## (c) Morphological correlations of androchromic variation: a test of pleiotropy

For 16 species, we assigned LD scores to each individual male and female, where higher LD scores were associated with more androchromic plumage (figure 2). Linear models revealed that higher LD scores in females showed no overall association with relative bill length across species. The direction of the associations varied among species (table 2 and figure 2). Only two species exhibited significant associations,

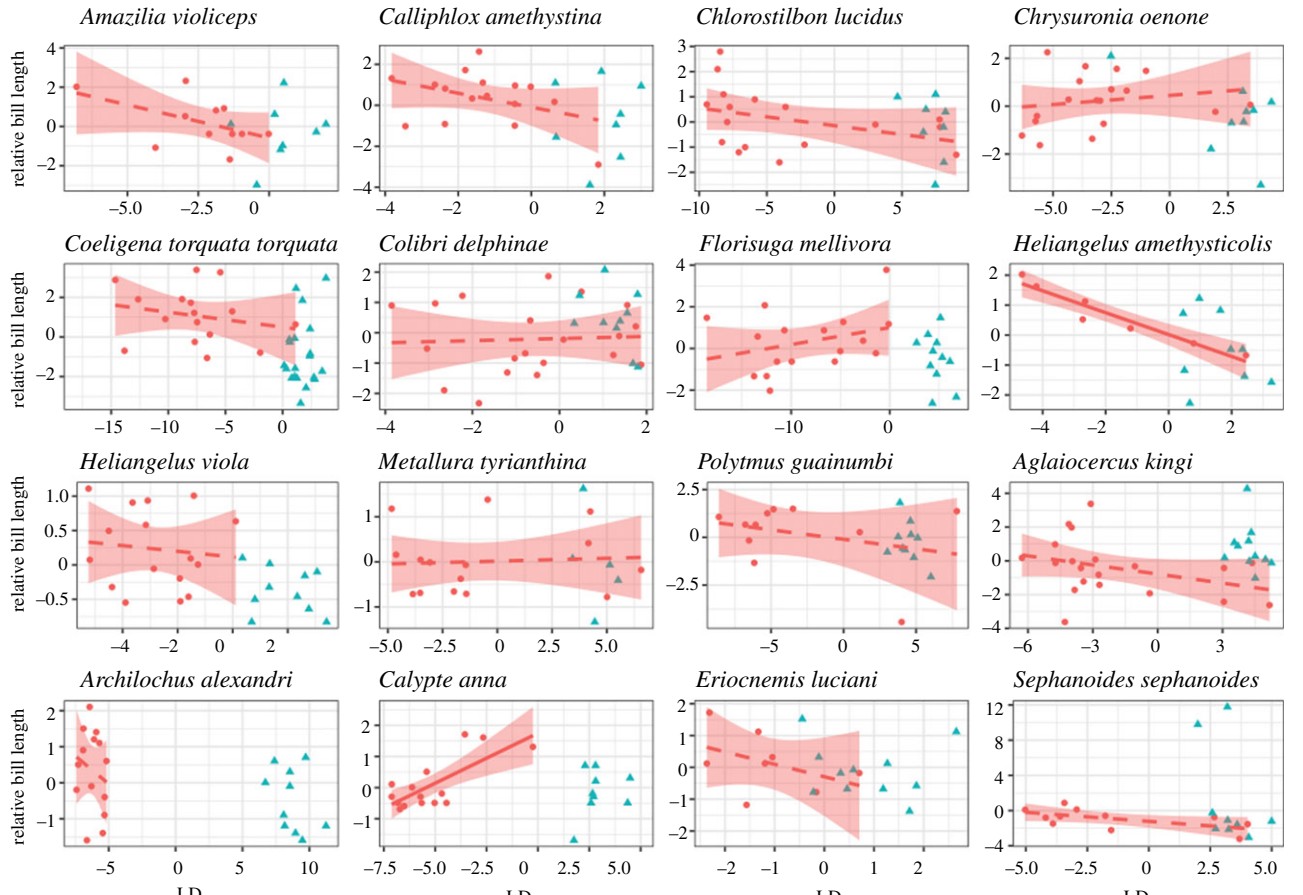

**Figure 2.** Association between relative bill length and androchromic plumage across tested species. Positive linear discriminant (LD) values are associated with more androchromic plumage. Negative LD is associated with more heterochromic plumage. Shaded regions represent 95% confidence intervals. Slopes are shown only for females (red circles) as the model is trained on androchromic males (blue triangles) and Class 1 heterochromic females. Solid regression lines denote significance ($p < 0.0031$). Dashed lines denote non-significance. (Online version in colour.)

the directionality of which differed between them. *Heliangelus amethysticolis* exhibited a significant negative correlation ($\beta = -2.53$; $p = 0.00058$) between relative bill length and LD1 (figure 2), whereas *Calypte anna* showed a positive correlation ($\beta = 1.71$; $p = 0.0016$) between bill length and LD1 (figure 2). *Heliangelus amethysticolis* was classified as exhibiting FLP in all of our analyses, whereas *Calypte anna* was classified as exhibiting FLP in some but not all of our analyses. There was no phylogenetic relationship between relative bill length and androchromic variation ($K = 0.34$; $p = 0.94$). There was also no general pattern of association between other morphological traits and androchromic variation in females across most species, though some species did exhibit significant trait associations (electronic supplementary material, text S8, table S2, figures S15–S16).

### (d) Trait associations of female-limited polymorphism: a comparative test of adaptive function

When comparing the binary character 'FLP' to 'no FLP' for all species, we found no significant effect of body size ($\beta = 0.088$, $p = 0.47$), social dominance ($\beta = 0.77$, $p = 0.18$) or migratory status ($\beta = -0.31$, $p = 0.10$). However, when excluding monochromic species, species were less likely to have FLP if they were migrants ($\beta = -0.35$, $p = 0.024$), and there was a trend for behaviourally dominant species to exhibit FLP over non-dominant species ($\beta = 0.87$, $P = 0.081$). There was no relationship with body size and FLP ($\beta = 0.057$, $p = 0.55$). Finally, we

found a slight but significant negative association with mean precipitation ($\beta = -0.008$, $p = 0.007$), but no other significant climate associations, when excluding monochromic species (electronic supplementary material, text S9).

## 4. Discussion

Previous work has demonstrated that multiple hummingbird species exhibit FLP where some females look like males [16,28,29]. We found that that roughly 25% of the nearly 200 species classified, including over 40% of dichromic species, exhibit FLP across the hummingbird lineage. These estimates are conservative, only categorizing species with bimodal plumage distributions across a heterochromic to androchromic continuum, while also controlling for risks of mis-sexing. Furthermore, this trait has likely evolved independently at least once in each hummingbird clade (for a total of 28 times), indicating its evolutionary lability and pervasiveness across the family. The existence and pervasiveness of FLP in which some females exhibit androchromic plumage challenges our understanding of the evolution of sexually dimorphic plumage traits in a system that has become a model for studying sexual selection [3,4,30].

To determine why FLP is so common in hummingbirds, we began by exploring two non-adaptive hypotheses. First, we tested if FLP was a non-adaptive byproduct of sexually antagonistic intralocus conflict where females resemble males due to strong selection on males and a shared genetic

**Table 2.** Morphological correlations between the degree of androchromic plumage (LD1) and relative bill length (bill length minus species' average bill length). Bolding denotes significant *p*-values.

| clade | species | bill length by LD1 slope | bill length by LD1 *p*-value |
|---|---|---|---|
| Topaz | *Florisuga mellivora*[a] | 1.18 | 0.226 |
| Mango | *Colibri delphinae*[a] | 0.077 | 0.829 |
| | *Polytmus guainumbi*[a] | −0.865 | 0.381 |
| Brilliant | *Aglaeactis pamela*[a] | −0.87 | 0.69 |
| | *Coeligena torquata torquata*[a] | −0.48 | 0.49 |
| | *Eriocnemis luciani*[c] | −0.47 | 0.29 |
| Coquette | *Heliangelus amethysticolis*[a] | **−2.53** | **<0.001** |
| | *Heliangelus viola*[a] | −0.30 | 0.68 |
| | *Metallura tyrianthina*[a] | 0.34 | 0.82 |
| | *Aglaeiocercus kingi*[b] | −0.68 | 0.13 |
| Bee | *Calliphlox amethystina*[a] | −0.94 | 0.41 |
| | *Archilochus alexandri*[d] | −0.70 | 0.075 |
| | *Calypte anna*[b] | **1.71** | **0.0016** |
| Emerald | *Chlorostilbon lucidus*[a] | −1.95 | 0.15 |
| | *Chrysuronia oenone*[a] | 0.410 | 0.48 |
| | *Amazilia violiceps*[c] | −0.74 | 0.12 |

[a]Species that were classified as exhibiting female-limited polymorphism (FLP) in all of our analyses.
[b]Species that were classified as exhibiting FLP in at least one, but not all, of our analyses.
[c]Species that were unable to be classified in our analyses.
[d]Species that were classified as not exhibiting FLP in any of our analyses.

architecture between the sexes [26,27]. According to this hypothesis, we would expect higher transition rates from sexual monochromic to FLP and from FLP to sexual dichromic, following the resolution of intralocus sexual conflict [27]. Instead, we found equal evolutionary transition rates across these plumage states, including in three sensitivity analyses. Three of the seven other reconstructions based on more liberal and conservative classification criteria produced symmetrical transition rates, with higher transition rates to and from sexual dichromic and FLP. Only one of the seven other reconstructions produced higher transition rates from sexual dichromic to FLP (electronic supplementary material, figure S14). In combination with the bimodality of female plumage variation in nearly 25% of all hummingbird species sampled, these results allow us to reject the non-adaptive hypothesis of sexually antagonistic intralocus conflict across hummingbirds. Of note, we were unable to define monochromic species as 'cryptic' and 'ornamented', as traits categorizing monomorphic species as 'ornamented' in other groups (e.g. not brown, exhibited spots or stripes, exhibiting iridescent patches, etc. [64]) describe all hummingbird species, including 'dull' females of sexually dimorphic species and monomorphic species [32]. Nonetheless, the number of sexually monochromic species that have been studied for their ornamentation is low and only constitutes two species [65,66]. Unlike the correlations in male and female tail lengths of Bee hummingbirds [67], the bimodality of androchromic plumage and the phylogenetic transition rates suggest that FLP in hummingbirds is not driven by sexual selection on males and intralocus sexual conflict.

Next, we tested the non-adaptive pleiotropy hypothesis whereby androchromic plumage is a consequence of indirect selection on other andromorphic traits through pleiotropy [12]. In hummingbirds, bill length is often sexually dimorphic and has implications for the types of flowers that each sex may be able to access [35,68]. Since sexually dimorphic traits are known to influence access to resources in hummingbirds [35,36], selection for male bill size, which could be linked to plumage in males, could indirectly select for androchromic plumage in females through pleiotropy [36]. To test this idea, we analysed the relationship between andromorphic bills and androchromic plumage across a range of species. Since we found that bills do not strongly differ in length across the range of heterochromic to androchromic female variation within species, and that there is no phylogenetic signal in these traits, we ruled out the non-adaptive pleiotropy hypothesis as it pertains to bill size. We also found no patterns of correlation between androchromic variation and wing length across all tested species (electronic supplementary material, figure S16). However, a few species did exhibit significant or marginally significant relationships between androchromic variation and morphology, suggesting that pleiotropy could play a limited role in androchromic variation in some hummingbird species. Although correlations with other unmeasured morphological traits could exist, we focused on bill length because it is the only sexually dimorphic trait that has been previously shown to have consequences for ecological niche breadth in hummingbirds [36].

Although non-adaptive pleiotropy may play a limited role in the evolution of FLP within individual species,

non-adaptive hypotheses do not fully explain the evolution and maintenance of FLP in hummingbirds generally. We explored potential socioecological factors that might provide preliminary evidence for an adaptive function of male-like coloration in females. We found significant associations with migratory status, lower mean precipitation, and marginal association of FLP with social dominance, all of which are linked to interspecific interactions and competition over resources [69–71]. Social dominance can directly mediate access to nectar, and differences in rainfall-mediated resource availability can impact intra- and interspecific competition [72], including among seasonal migrants [34,73]. However, since sex separation in migration could also decrease competition [61–63], further work is needed to understand how migration might influence competition and ultimately plumage evolution in hummingbirds. Indeed, interspecific— and sometimes intersexual—competition is common among hummingbirds [34,74–76], and these socioecological factors may promote the evolution of bright plumage ornamentation as status symbols to mediate such social conflict [65,66]. Further empirical work is needed to decipher how inter- and intraspecific competition varies with FLP within and across species' ranges, and how adaptive hypotheses may play a role in the evolution of FLP.

## 5. Conclusion

In summary, our results indicate that FLP is widespread across the hummingbird lineage and has evolved independently multiple times. We rejected non-adaptive hypotheses for FLP by analysing evolutionary transition rates, confirming the bimodal distribution of androchromic plumage within hummingbird species, and finding no association between androchromic plumage and andromorphic morphology. Instead, we found preliminary support for the idea that socioecological traits best predict the evolution of FLP across the phylogeny, making social selection a potential explanation for the evolution and maintenance of this trait. Hummingbird plumage ornaments are classic examples of sexual selection, but these traits may have a more complex function than previously considered. Gaining a better understanding of female signalling traits in a group that exhibits widespread variation in female plumage within and across species will enable us to test hypotheses explaining female ornamentation, which are quite distinct from those underlying ideas of classic sexual selection on male ornaments [7,8,20]. By considering variation in female coloration across a family of highly ornamented birds, we demonstrate that non-adaptive explanations are unlikely to maintain a surprisingly common polymorphism and female ornamentation more generally. Instead, our comparative analyses suggest that socioecological hypotheses should be investigated as a potential alternative for the evolution and maintenance of ornamentation in female birds.

Data accessibility. The datasets supporting this article are available from the Dryad Digital Repository: https://doi.org/10.5068/D1XD43 [77].

Authors' contributions. E.S.D. carried out the data collection, statistical analyses, participated in the design of the study and drafted the manuscript; J.J.F. participated in the design of the study, participated in the statistical analyses and critically revised the manuscript; D.R.R. conceived of the study, designed the study, coordinated the study and critically revised the manuscript. All authors gave final approval for publication and agree to be held accountable for the work performed therein.

Competing Interests. We declare we have no competing interests.

Funding. E.S.D. was funded by the Wilson Ornithological Society and the Department of Ecology, Evolution and Environmental Biology at Columbia University.

Acknowledgements. We acknowledge Rafael Maia and Joel Cracraft for constructive feedback. We are grateful to Jim McGuire for providing access to the Trochilidae phylogeny, Myron Huang, Michael Spiotta, Jake Arden, Francesca Garofalo, John McCormack (Moore Lab of Zoology), and James Maley (Moore Lab of Zoology) for helping collect data, and Kevin Schwarzwald, Mark Juhn and Carlos Botero for helping with statistical methods. We thank three anonymous reviewers for constructive critique. This work would not have been possible without being granted access by the curators and collection managers of: The American Museum of Natural History (Joel Cracraft and Paul Sweet); the LSU Museum of Science (J.V. Remsen and Steve Cardiff); Smithsonian National Museum of Natural History (Gary Graves, Helen Jeames and Chris Milensky); The Field Museum (Shannon Hacket, John Bates and Ben Marks); and the Los Angeles Museum of Natural History (Kenneth Campbell and Kimball Garrett).

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
