## [Peer Review File · Proceedings of the Royal Society B: Biological Sciences]

Review History

RSPB-2020-1963.R0 (Original submission)

Review form: Reviewer 1 (William Allen)

Recommendation

Major revision is needed (please make suggestions in comments)

Scientific importance: Is the manuscript an original and important contribution to its field?

Good

General interest: Is the paper of sufficient general interest?

Good

Quality of the paper: Is the overall quality of the paper suitable?

Good

Is the length of the paper justified?

Yes

Should the paper be seen by a specialist statistical reviewer?

No

Do you have any concerns about statistical analyses in this paper? If so, please specify them explicitly in your report.

No

It is a condition of publication that authors make their supporting data, code and materials available - either as supplementary material or hosted in an external repository. Please rate, if applicable, the supporting data on the following criteria.

Is it accessible?

Yes

Is it clear?

Yes

Is it adequate?

Yes

Do you have any ethical concerns with this paper?

No

Comments to the Author

Dear Authors,

I enjoyed reading your paper. It is an interesting subject for all the reasons you describe. In general I think the investigation has been well conducted and reported but there are several areas where improvements could straightforwardly be made. I will preface my detailed comments by saying I am not particularly knowledgeable about hummingbird ecology and evolution, so some of my points might reflect my own lack of understanding, in which case they could be resolved with inclusion of more natural history and background in the text.

While it is always difficult to draw conclusions from null results and the predictions (L105-116) for the non-adaptive hypotheses could be described more clearly, I think they are good tests of intralocus conflict and pleiotropy hypotheses. My only real query on this aspect was whether there is dimorphism in traits other than bill size that might be involved in pleiotropic effects on F plumage?

The investigation of the adaptive hypothesis is much more exploratory. This is fine in principle if it is flagged as such throughout, but this only happens in some places and not others. In the introduction there is no explanation for why the specific socioecological variables were selected, and what relationships might be expected under adaptive hypotheses. However in the abstract and discussion weak associations between FLP and migration and social dominance are taken to support a "social benefits" explanation of FLP. This seems very preliminary. In my view, the conclusion "we were able to demonstrate that social selection maintains ... a female limited polymorphism" is not clearly supported by the presented evidence. In particular I am concerned that this evidence rests on the decision taken at the end of the methods to exclude monochromatic species. This is not explained in the text. As this is the only analysis that returns (marginally) significant results, the reader will want to know why the decision was taken to explore only this subset of data. What theoretical justification is there for removing monochromatic species? Were any other subsets also explored? Was the decision to explore the subset taken before or after results including the monochromatic species were known? There is a fine line between exploring a dataset and dredging, so I think the MS needs to be very clear about the analysis decisions. Related to this point is the choice of behavioural and morphological traits to test the adaptive hypothesis. If the social signalling hypothesis is the prime adaptive hypothesis then why measure migratory tendency and not a variable more directly related to territoriality and social behaviour? Other traits that might be more appropriate that might be able to be analysed in a revised version of this paper include position in interspecific dominance hierarchy (or does this always depend

on body size?) and the number or similarity of sympatric species. In general, I would like to see more explanation and context given to how the adaptive hypothesis is investigated, and the results of this analysis. At present, while I appreciate you are clear more work needs to be done on the social signalling hypothesis, the current explanation for why a fairly weak association with migratory tendency supports it seems very convoluted and tenuous. It is possible to think of other explanations for an association of FLP with migration (not forgetting this is only among dichromatic species), for example disruptive selection on F plumage arising from different selection pressures in summer and winter ranges.

There are other places in the introduction too where I would like to see arguments better developed to help the reader follow including L59-60 – why might it provide insights?; L87-90 why does this work suggest adaptation but not rule out non-adaptive explanations?; L97 why is determining evolutionary history necessary for inferring function?

The other part I had difficulty following was in quantifying FLP. Being unfamiliar with Bleiweiss I found description of this confusing. The text and SI should stand alone. Specimens both differ in the number of androchromic patches and the area of each patch with androchromic colour and this determines % score – correct? Figure S1 didn't especially help – something like Fig S4 might be more useful if the same patches were used? I also wondered why convert the % to a 4-level ordinal variable. Wouldn't it be better to do uni/bimodal classification on the continuous scores? Finally, without methodological detail on how the “whole body” image analysis was conducted, its inclusion doesn't address the issue that subjective scores might quantify variation as well as a more objective analysis (the objective analysis could just be done badly). I suggest removing it. Scoring discreet patches subjectively is perfectly appropriate given this study's goals. If it stays then it must be properly reported.

Another aspect that it might be possible to straightforwardly address is handling of phylogenetic uncertainty. In ancestral state reconstructions and identification of the number of independent origins of a trait the tree topology is very critical – if a tree block were available it would be great to run analyses across this to get bounds on estimates.

I hope these comments are useful in improving the manuscript. I also make a few minor suggestions below.

Yours sincerely,
Will Allen

L48 -50 Does this sentence intentionally ignore selection consequence on male preferences, or is it just unclear? My understanding is that theory and evidence currently point to the possibility of M choice leading to the evolution of F signals in polygynous species.

L67 Inconsistent reference formatting.

L99 I suggest writing this paragraph in the present tense.

Specimens: The size of the sample is very impressive. I think it would be useful to present a histogram of species' sample sizes in the SI.

L167 I think it is Hartigan's test.

L177 Novel species or increasing the sample size of species already in the dataset?

L219 I disagree with the decision to fix the probabilities at the root. There are many evolutionary scenarios where FLP could have been present at the root. Why not just let the data decide?

L232 “reconstruct trait evolution on trees”

L249 Report the mean sample size of the three groups not the target sample size.

L261 It is a shame that a UV sensitive camera was not used, as hummingbirds are UV sensitive. This issue (that specimens may differ in ways invisible to the camera and experimenter, but visible to hummingbirds) should be acknowledged in the text.

L282 aggressive towards conspecifics, heterospecifics or both?

Table S1 suggests that when a liberal 'yes' threshold is used the ARD model has a lower AICc score. This is not what is reported in the text. Is this an error or have I misunderstood?

L327 Is 2/16 significant associations beyond what would be expected by chance (multiple comparison problem)?

L357 I assume all monochromatic species all have 'female like' cryptic colour but this should perhaps be pointed out as if all or some have male like colour the predictions would change.

Review form: Reviewer 2 (Hugo Gruson)

Recommendation

Accept with minor revision (please list in comments)

Scientific importance: Is the manuscript an original and important contribution to its field?

Excellent

General interest: Is the paper of sufficient general interest?

Excellent

Quality of the paper: Is the overall quality of the paper suitable?

Excellent

Is the length of the paper justified?

Yes

Should the paper be seen by a specialist statistical reviewer?

No

Do you have any concerns about statistical analyses in this paper? If so, please specify them explicitly in your report.

No

It is a condition of publication that authors make their supporting data, code and materials available - either as supplementary material or hosted in an external repository. Please rate, if applicable, the supporting data on the following criteria.

Is it accessible?

Yes

Is it clear?

No

Is it adequate?

Yes

Do you have any ethical concerns with this paper?

No

Comments to the Author

(this markdown-formatted review is also available in compiled format as an attachment)

Review for manuscript 'Male-like female morphs in hummingbirds: the evolution of a widespread sex-limited plumage polymorphism' (for ProcB)

In this manuscript, the authors address the question of the existence of sex-limited polymorphism with male-like females (in terms of plumage colouration) in hummingbirds. This topic has been discussed multiple times at the intraspecific level, especially by Bleiweiss, but surprisingly, no study had yet approached it from the interspecific angle, by looking at the whole hummingbird family. Here, the authors use the fairly recent advances in the reconstruction of the phylogeny of the family and gather an extensive dataset from various museums to try and find out which evolutionary mechanisms drive the evolution of female-limited polymorphism.

Overall, I think the manuscript is of good quality. The science seems sound and it reads well. Importantly, it does fit quite well with ProcB format. I think some specific points could be further clarified, and maybe sometimes require additional/different analyses but I'm willing to be convinced otherwise.

Line by line comments

- L87-88:

> "Together, this work suggests an adaptive function of female androchromic plumage rather than a non-adaptive genetic correlation with males"

I think you're being too fast here. As you mention in the following sentences, this is compatible with the pleiotropy hypothesis, which is not adaptative. Something like "Together, this work tends to disprove the sexually antagonistic intralocus conflict hypothesis." This also has the advantage of repeating verbatim the name of one of your hypothesis, which helps the reader memorizing the various alternatives.

- L106-108:

> "(2) higher transition rates from sexual monomorphism to FLP and from FLP to sexual dimorphism than from sexual dimorphism to FLP or from FLP to sexual monomorphism."

I didn't really understand the reason for this at first and it's only later in the manuscript (L359) that I understood. Your explanation in the discussion is better in my opinion and the mention of the "resolution of intralocus sexual conflict" is absolutely crucial to fully understand your predictions.

- L132-133:

> "including those in which sexing methods were unknown (63% of samples)"

Since this is a weak point of your study, it may be worthwhile to mention in this 'Sexing' subsection that you try to take into account this uncertainty in your analyses (by using different thresholds).

- L141:

> "those with unknown sexing methods across sexually dimorphic species for which there were ≥ 10 females with unknown sexing, ≥ 10 females with gonad data, and at least one androchrome observed (N = 41 species)."

By using the data you provided, I find $N = 32$ (see attached script). It may be because I misunderstood your dataset so please double check and either correct this number or make your data more re-usable (by using more machine-friendly column names and by providing a key explaining the meaning of each column).

- L149:

> "by one researcher"

Given the size of the dataset, I understand it was not possible to have multiple people to rate the specimens. But since I'm assuming the first author of this study did the observation, has you given a thought about how your preconceived explanations for FLP might have influenced your observations?

- L151-153:

In the supplementary data that you provided, what does "1,4" mean in the `Class` column?

- L269-270:

> "We normalized bill length by dividing it by wing length to account for body size differences among species."

I don't think wing length is a good body size index in hummingbirds. Why not use the full body size? You seem to have this data since you mention it in the next subsection. I'm not sure scaling is necessary at all anyways. I'd instead present "bill length by LD1" that you included in ESM.

- SI L40-41:

> "We extracted the latitude and longitude of the centroid of the range of each species using range maps."

This doesn't seem like the most robust way to do this. For starters, how did you do with disjoint ranges? The centroid might land in an area where the species is absent. Additionally, even if the range does not have disjoint components, the centroid may be in the most extreme area in terms of climate. A much more robust option in my opinion would be to extract climatic data from the whole range, and *then* take the mean/median.

Decision letter (RSPB-2020-1963.R0)

07-Sep-2020

Dear Ms Diamant:

We have now received two reviews of your manuscript RSPB-2020-1963 entitled "Male-like female morphs in hummingbirds: the evolution of a widespread sex-limited plumage polymorphism." Both reviewers, the Associate Editor, and I are very enthusiastic about many aspects of your manuscript, however the reviewers and the AE have all recommended that substantial revisions are necessary. In particular, one reviewer, the AE, and I would like to see stronger evidence in favor of the adaptive explanation if that is to be the one supported. With this in mind we would be happy to consider a resubmission, provided the comments of the referees are fully addressed. Please note that this is not a provisional acceptance.

The resubmission will be treated as a new manuscript. However, we will approach the same reviewers if they are available and it is deemed appropriate to do so by the Editor. Please note that resubmissions must be submitted within six months of the date of this email. In exceptional

circumstances, extensions may be possible if agreed with the Editorial Office. Manuscripts submitted after this date will be automatically rejected.

Sincerely,

Dr Sarah Brosnan
 Editor, Proceedings B
 mailto: proceedingsb@royalsociety.org

Associate Editor

Comments to Author:

The manuscript has been reviewed by two experts in the field. Both reviewers like the topic and fit for the Journal. While Reviewer 2 recommends that only minor revisions are necessary, Reviewer 1 finds that there are significant issues with the analyses that lead to the overstating of the conclusions regarding the adaptive value of the female-linked polymorphisms. I agree with the Reviewer that stronger evidence is needed before the adaptive interpretation would be supported. I would like to give you the opportunity to respond to these concerns. However, please note that we are hoping for a substantive effort to address these concerns, and that there is no guarantee of eventual acceptance of this manuscript. If you choose to send us a revised manuscript, it will be sent back to the original reviewers, should they be available and willing to view a revision.

In addition to the comments of both reviewers, I would add that I felt that the structure of the manuscript could be clearer. There seem to be 4 aims given in the last paragraph of the introduction, but they are buried and hard to pull out. These do map on to 4 results subsections, but the language used in the introduction and the results section to describe these is not the same. Similar structures and headings are then used in the methods and results sections, yet they do differ, which makes the manuscript unnecessarily hard to follow. For example, a Methods subheading "Morphological correlations of androchromic variation within and across species", followed by a Results subheading: "Morphological correlations of androchromic variation: a test of pleiotropy". It would be much easier to track the manuscript if exactly the same structure, subheaded by exactly the same (not just similar) words were used in each section for these 4 aims (starting in the introduction) wherever possible.

Reviewer(s)' Comments to Author:

Referee: 1

Comments to the Author(s)

Dear Authors,

I enjoyed reading your paper. It is an interesting subject for all the reasons you describe. In general I think the investigation has been well conducted and reported but there are several areas where improvements could straightforwardly be made. I will preface my detailed comments by saying I am not particularly knowledgeable about hummingbird ecology and evolution, so some of my points might reflect my own lack of understanding, in which case they could be resolved with inclusion of more natural history and background in the text.

While it is always difficult to draw conclusions from null results and the predictions (L105-116) for the non-adaptive hypotheses could be described more clearly, I think they are good tests of intralocus conflict and pleiotropy hypotheses. My only real query on this aspect was whether there is dimorphism in traits other than bill size that might be involved in pleiotropic effects on F plumage?

The investigation of the adaptive hypothesis is much more exploratory. This is fine in principle if it is flagged as such throughout, but this only happens in some places and not others. In the introduction there is no explanation for why the specific socioecological variables were selected, and what relationships might be expected under adaptive hypotheses. However in the abstract and discussion weak associations between FLP and migration and social dominance are taken to support a “social benefits” explanation of FLP. This seems very preliminary. In my view, the conclusion “we were able to demonstrate that social selection maintains ... a female limited polymorphism” is not clearly supported by the presented evidence. In particular I am concerned that this evidence rests on the decision taken at the end of the methods to exclude monochromatic species. This is not explained in the text. As this is the only analysis that returns (marginally) significant results, the reader will want to know why the decision was taken to explore only this subset of data. What theoretical justification is there for removing monochromatic species? Were any other subsets also explored? Was the decision to explore the subset taken before or after results including the monochromatic species were known? There is a fine line between exploring a dataset and dredging, so I think the MS needs to be very clear about the analysis decisions. Related to this point is the choice of behavioural and morphological traits to test the adaptive hypothesis. If the social signalling hypothesis is the prime adaptive hypothesis then why measure migratory tendency and not a variable more directly related to territoriality and social behaviour? Other traits that might be more appropriate that might be able to be analysed in a revised version of this paper include position in interspecific dominance hierarchy (or does this always depend on body size?) and the number or similarity of sympatric species. In general, I would like to see more explanation and context given to how the adaptive hypothesis is investigated, and the results of this analysis. At present, while I appreciate you are clear more work needs to be done on the social signalling hypothesis, the current explanation for why a fairly weak association with migratory tendency supports it seems very convoluted and tenuous. It is possible to think of other explanations for an association of FLP with migration (not forgetting this is only among dichromatic species), for example disruptive selection on F plumage arising from different selection pressures in summer and winter ranges.

There are other places in the introduction too where I would like to see arguments better developed to help the reader follow including L59-60 – why might it provide insights?; L87-90 why does this work suggest adaptation but not rule out non-adaptive explanations?; L97 why is determining evolutionary history necessary for inferring function?

The other part I had difficulty following was in quantifying FLP. Being unfamiliar with Bleiweiss I found description of this confusing. The text and SI should stand alone. Specimens both differ in the number of androchromic patches and the area of each patch with androchromic colour and this determines % score – correct? Figure S1 didn’t especially help – something like Fig S4 might be more useful if the same patches were used? I also wondered why convert the % to a 4-level ordinal variable. Wouldn’t it be better to do uni/bimodal classification on the continuous scores? Finally, without methodological detail on how the “whole body” image analysis was conducted, its inclusion doesn’t address the issue that subjective scores might quantify variation as well as a

more objective analysis (the objective analysis could just be done badly). I suggest removing it. Scoring discreet patches subjectively is perfectly appropriate given this study's goals. If it stays then it must be properly reported.

Another aspect that it might be possible to straightforwardly address is handling of phylogenetic uncertainty. In ancestral state reconstructions and identification of the number of independent origins of a trait the tree topology is very critical – if a tree block were available it would be great to run analyses across this to get bounds on estimates.

I hope these comments are useful in improving the manuscript. I also make a few minor suggestions below.

Yours sincerely,

Will Allen

L48 -50 Does this sentence intentionally ignore selection consequence on male preferences, or is it just unclear? My understanding is that theory and evidence currently point to the possibility of M choice leading to the evolution of F signals in polygynous species.

L67 Inconsistent reference formatting.

L99 I suggest writing this paragraph in the present tense.

Specimens: The size of the sample is very impressive. I think it would be useful to present a histogram of species' sample sizes in the SI.

L167 I think it is Hartigan's test.

L177 Novel species or increasing the sample size of species already in the dataset?

L219 I disagree with the decision to fix the probabilities at the root. There are many evolutionary scenarios where FLP could have been present at the root. Why not just let the data decide?

L232 "reconstruct trait evolution on trees"

L249 Report the mean sample size of the three groups not the target sample size.

L261 It is a shame that a UV sensitive camera was not used, as hummingbirds are UV sensitive. This issue (that specimens may differ in ways invisible to the camera and experimenter, but visible to hummingbirds) should be acknowledged in the text.

L282 aggressive towards conspecifics, heterospecifics or both?

Table S1 suggests that when a liberal 'yes' threshold is used the ARD model has a lower AICc score. This is not what is reported in the text. Is this an error or have I misunderstood?

L327 Is 2/16 significant associations beyond what would be expected by chance (multiple comparison problem)?

L357 I assume all monochromatic species all have 'female like' cryptic colour but this should perhaps be pointed out as if all or some have male like colour the predictions would change.

Referee: 2

Comments to the Author(s)

(this markdown-formatted review is also available in compiled format as an attachment)

Review for manuscript 'Male-like female morphs in hummingbirds: the evolution of a widespread sex-limited plumage polymorphism' (for ProcB)

In this manuscript, the authors address the question of the existence of sex-limited polymorphism with male-like females (in terms of plumage colouration) in hummingbirds. This topic has been discussed multiple times at the intraspecific level, especially by Bleiweiss, but surprisingly, no study had yet approached it from the interspecific angle, by looking at the whole hummingbird family. Here, the authors use the fairly recent advances in the reconstruction of the phylogeny of the family and gather an extensive dataset from various museums to try and find out which evolutionary mechanisms drive the evolution of female-limited polymorphism.

Overall, I think the manuscript is of good quality. The science seems sound and it reads well. Importantly, it does fit quite well with ProcB format. I think some specific points could be further clarified, and maybe sometimes require additional/different analyses but I'm willing to be convinced otherwise.

Line by line comments

- L87-88:

> "Together, this work suggests an adaptive function of female androchromic plumage rather than a non-adaptive genetic correlation with males"

I think you're being too fast here. As you mention in the following sentences, this is compatible with the pleiotropy hypothesis, which is not adaptive. Something like "Together, this work tends to disprove the sexually antagonistic intralocus conflict hypothesis." This also has the advantage of repeating verbatim the name of one of your hypothesis, which helps the reader memorizing the various alternatives.

- L106-108:

> "(2) higher transition rates from sexual monomorphism to FLP and from FLP to sexual dimorphism than from sexual dimorphism to FLP or from FLP to sexual monomorphism."

I didn't really understand the reason for this at first and it's only later in the manuscript (L359) that I understood. Your explanation in the discussion is better in my opinion and the mention of the "resolution of intralocus sexual conflict" is absolutely crucial to fully understand your predictions.

- L132-133:

> "including those in which sexing methods were unknown (63% of samples)"

Since this is a weak point of your study, it may be worthwhile to mention in this 'Sexing' subsection that you try to take into account this uncertainty in your analyses (by using different thresholds).

- L141:

> "those with unknown sexing methods across sexually dimorphic species for which there were ≥ 10 females with unknown sexing, ≥ 10 females with gonad data, and at least one androchrome observed (N = 41 species)."

By using the data you provided, I find N = 32 (see attached script). It may be because I misunderstood your dataset so please double check and either correct this number or make your data more re-usable (by using more machine-friendly column names and by providing a key explaining the meaning of each column).

- L149:

> "by one researcher"

Given the size of the dataset, I understand it was not possible to have multiple people to rate the specimens. But since I'm assuming the first author of this study did the observation, has you given a thought about how your preconceived explanations for FLP might have influenced your observations?

- L151-153:

In the supplementary data that you provided, what does "1,4" mean in the `Class` column?

- L269-270:

> "We normalized bill length by dividing it by wing length to account for body size differences among species."

I don't think wing length is a good body size index in hummingbirds. Why not use the full body size? You seem to have this data since you mention it in the next subsection. I'm not sure scaling is necessary at all anyways. I'd instead present "bill length by LD1" that you included in ESM.

- SI L40-41:

> "We extracted the latitude and longitude of the centroid of the range of each species using range maps."

This doesn't seem like the most robust way to do this. For starters, how did you deal with disjoint ranges? The centroid might land in an area where the species is absent. Additionally, even if the range does not have disjoint components, the centroid may be in the most extreme area in terms of climate. A much more robust option in my opinion would be to extract climatic data from the whole range, and *then* take the mean/median.

Author's Response to Decision Letter for (RSPB-2020-1963.R0)

See Appendix A.

RSPB-2020-3004.R1 (Revision)

Review form: Reviewer 3

Recommendation

Major revision is needed (please make suggestions in comments)

Scientific importance: Is the manuscript an original and important contribution to its field?

Excellent

General interest: Is the paper of sufficient general interest?

Good

Quality of the paper: Is the overall quality of the paper suitable?

Acceptable

Is the length of the paper justified?

Yes

Should the paper be seen by a specialist statistical reviewer?

No

Do you have any concerns about statistical analyses in this paper? If so, please specify them explicitly in your report.

Yes

It is a condition of publication that authors make their supporting data, code and materials available - either as supplementary material or hosted in an external repository. Please rate, if applicable, the supporting data on the following criteria.

Is it accessible?

No

Is it clear?

N/A

Is it adequate?

N/A

Do you have any ethical concerns with this paper?

No

Comments to the Author

In this study, the authors tested how widespread FLP was in hummingbirds and then tested what might predict the distribution and evolution of this trait. They found that FLP was widespread in hummingbirds and likely evolved independently in each major tribe. They then tested the non-adaptive hypotheses of female colouration being driven by sexual selection on males or pleiotropy. They concluded that these hypotheses were not supported. Then the tested three ecological variables to see if these traits predicted FLP evolution and found support for migration and climate to predict FLP evolution.

Overall, I found this manuscript to be interesting and on a topic of importance. I applaud the authors for the breadth of their examination and testing of many different hypotheses in one study. I think this paper has great potential and could be a strong contribution to Proceedings B.

However, I have some reservations about the analyses and interpretation of the results that should be addressed. I also had several questions about the methods that should be clarified to help with the interpretation of the results.

First, I am not fully convinced that sexually antagonistic intralocus conflict hypothesis has been rejected by the current analyses. Specifically, I'm not sure if the results from Lines 330-338 & Lines 378-397 alone are fully convincing that the equal rates model demonstrates that FLP is not a transition state. Did the authors consider using a symmetrical rates model (SYM), as the ARD model can be heavily penalized through AIC due to the many different estimates in the model. I think the authors need to consider more evolutionary models than the equal rates and all rates different models and should also look at the number of transitions between each of the different states over time. Perhaps utilize similar analyses to Ligon et al. 2016 (*Evolution*, 70: 2839-2852).

Secondly, the statements from Lines 351-352 & 406-410 seem mis-leading to me, unless I am reading the results from the ESM wrong, as it seems there were quite a few correlations between wing length/bill length with LD1. Even if these ESM results are marginally significant under the corrected p-values they must be interpreted, since the authors are also interpreting a marginally significant result with regards to FLP and social dominance (lines 418-419).

Thirdly, I do not think I understand how relationships with migration and precipitation indicate that social interactions and territoriality are driving the evolution of FLP (Lines 419-423). These are interesting relationships that are definitely worth exploring in the discussion, but the current discussion of these results seems lacking and shoe-horned to fit a specific social selection explanation. Also, what were the sample sizes for the ecological analyses? If they used the larger dataset, then how can the authors reconcile the differences in sample sizes between the LD1 analyses and these analyses?

I had the following addition comments as well:

Lines 114-115: Would you not also expect this relationship in males? Because if there was no relationship between male colour and bill morphology, but there was a relationship between female androchromatic colour and female andromorphic bill morphology, that would not necessarily support pleiotropy, in my mind.

Lines 150-152: Perhaps include in parentheses the initials of which author or researcher scaled the plumage?

Lines 152-155: was this classification based on differences in hue and patch size alone? Or were differences in brightness, chroma, and angle-dependence also considered? More details on what plumage features were used to quantify differences between males and females would be helpful.

Lines 163: does this mean that degree of androchromatic plumage was only classified based on their iridescent plumage? Or was the whole body used and this reference to iridescent plumage is only about determining which subspecies to use?

Lines 193-203: It would be help to mention that the authors also developed a conservative absence and liberal presence thresholds and that the descriptions for each are in the ESM. I was a bit confused in reading this section the first time through.

Lines 263-266: I think the authors should also acknowledge that the lack of UV reflectance measured is a limitation of their methods, instead of just writing it off as a non-issue. Especially since citation 53 is unpublished as well. Also see Burns & Shultz 2012 (*The Auk*, 2: 211-221)

Lines 258-260: Why would some species have additional photographs? What are some examples of these patches used?

Lines 268-278: I am confused about the purpose of the LDA here. The authors are comparing males to cryptic-morph females? Or male-morph females to cryptic-morph females? If the former, were male-morph females also included? And why were only Class 1 females used? I thought Class 1 and 2 were lumped together in prior analyses?

Lines 281-282: What is the sample size for this analysis? Is it the larger dataset or smaller again?

Lines 292-293: Were the climate variables also included as fixed effects in this model?

Lines 321-322: It is curious that FPL evolved twice in the Bee clade, when only 1 species exhibits FPL. Any thoughts on this?

Lines 340-352: The authors mention *Calypte anna* here and *Calliphlox amethystina* in the ESM but said that only 1 bee species exhibited FLP. Is this a typo?

ESM - It would help if the authors labeled the different textual parts of the ESM as Text S1, Text S2 etc., so that when they refer to the ESM in the main text, the reader knows exactly what part is referenced.

ESM - Sensitivity analyses: The difference between an 85% and 90% chance to detect an androchrome female does not seem that big of a difference to me, especially for one to be labeled as liberal and the other conservative. Is there a reason for using these numbers? Why not 95% or 99% for the conservative? Or 70% for the liberal?

ESM - Table S1: to better correspond with the text, I would change the "yes" and "no" classifications to "presence" and "absence"

Decision letter (RSPB-2020-3004.R0)

12-Jan-2021

Dear Ms Diamant:

Your manuscript has now been peer reviewed and the reviews have been assessed by an Associate Editor. As you will see, the reviewer (who is new since your last round of review) and the Associate Editor both find a lot to like about your manuscript, but the new reviewer has raised some additional concerns that need to be addressed the clarify and further strengthen your manuscript. The reviewer's comments (not including confidential comments to the Editor) and the comments from the Associate Editor are included at the end of this email for your reference.

If deemed necessary by the Associate Editor, your manuscript will be sent back to one or more of the original reviewers for assessment. If the original reviewers are not available we may invite new reviewers. Please note that we cannot guarantee eventual acceptance of your manuscript at this stage.

Research ethics:

Use of animals and field studies:

It is a condition of publication that you make available the data and research materials supporting the results in the article (<https://royalsociety.org/journals/authors/author-guidelines/#data>). Datasets should be deposited in an appropriate publicly available repository and details of the associated accession number, link or DOI to the datasets must be included in the Data Accessibility section of the article (<https://royalsociety.org/journals/ethics-policies/data-sharing-mining/>). Reference(s) to datasets should also be included in the reference list of the article with DOIs (where available).

Please submit a copy of your revised paper within three weeks. If we do not hear from you within this time your manuscript will be rejected. If you are unable to meet this deadline please let us know as soon as possible, as we may be able to grant a short extension.

Best wishes,
Dr Sarah Brosnan
Editor, Proceedings B
mailto:proceedingsb@royalsociety.org

Associate Editor Board Member

Comments to Author:

The manuscript has now been assessed by a further reviewer. This reviewer is (like the original two), also supportive of the manuscript, and thinks that it would make a strong addition to Proceedings B. The reviewer has made some suggestions about the modeling approach, and has also made a number of suggestions for additional clarifications. I think that these could usefully be addressed in a further revision. I am optimistic that the authors will be able to address all of these comments.

Reviewer(s)' Comments to Author:

Referee: 3

Comments to the Author(s).

In this study, the authors tested how widespread FLP was in hummingbirds and then tested what might predict the distribution and evolution of this trait. They found that FLP was widespread in hummingbirds and likely evolved independently in each major tribe. They then tested the non-adaptive hypotheses of female colouration being driven by sexual selection on males or pleiotropy. They concluded that these hypotheses were not supported. Then the tested three ecological variables to see if these traits predicted FLP evolution and found support for migration and climate to predict FLP evolution.

Overall, I found this manuscript to be interesting and on a topic of importance. I applaud the authors for the breadth of their examination and testing of many different hypotheses in one study. I think this paper has great potential and could be a strong contribution to Proceedings B. However, I have some reservations about the analyses and interpretation of the results that should be addressed. I also had several questions about the methods that should be clarified to help with the interpretation of the results.

First, I am not fully convinced that sexually antagonistic intralocus conflict hypothesis has been rejected by the current analyses. Specifically, I'm not sure if the results from Lines 330-338 & Lines 378-397 alone are fully convincing that the equal rates model demonstrates that FLP is not a transition state. Did the authors consider using a symmetrical rates model (SYM), as the ARD model can be heavily penalized through AIC due to the many different estimates in the model. I think the authors need to consider more evolutionary models than the equal rates and all rates different models and should also look at the number of transitions between each of the different states over time. Perhaps utilize similar analyses to Ligon et al. 2016 (*Evolution*, 70: 2839-2852).

Secondly, the statements from Lines 351-352 & 406-410 seem mis-leading to me, unless I am reading the results from the ESM wrong, as it seems there were quite a few correlations between wing length/bill length with LD1. Even if these ESM results are marginally significant under the corrected p-values they must be interpreted, since the authors are also interpreting a marginally significant result with regards to FLP and social dominance (lines 418-419).

Thirdly, I do not think I understand how relationships with migration and precipitation indicate that social interactions and territoriality are driving the evolution of FLP (Lines 419-423). These are interesting relationships that are definitely worth exploring in the discussion, but the current discussion of these results seems lacking and shoe-horned to fit a specific social selection explanation. Also, what were the sample sizes for the ecological analyses? If they used the larger dataset, then how can the authors reconcile the differences in sample sizes between the LD1 analyses and these analyses?

I had the following addition comments as well:

Lines 114-115: Would you not also expect this relationship in males? Because if there was no relationship between male colour and bill morphology, but there was a relationship between female androchromatic colour and female andromorphic bill morphology, that would not necessarily support pleiotropy, in my mind.

Lines 150-152: Perhaps include in parentheses the initials of which author or researcher scaled the plumage?

Lines 152-155: was this classification based on differences in hue and patch size alone? Or were differences in brightness, chroma, and angle-dependence also considered? More details on what plumage features were used to quantify differences between males and females would be helpful.

Lines 163: does this mean that degree of androchromatic plumage was only classified based on their iridescent plumage? Or was the whole body used and this reference to iridescent plumage is only about determining which subspecies to use?

Lines 193-203: It would be help to mention that the authors also developed a conservative absence and liberal presence thresholds and that the descriptions for each are in the ESM. I was a bit confused in reading this section the first time through.

Lines 263-266: I think the authors should also acknowledge that the lack of UV reflectance measured is a limitation of their methods, instead of just writing it off as a non-issue. Especially since citation 53 is unpublished as well. Also see Burns & Shultz 2012 (*The Auk*, 2: 211-221)

Lines 258-260: Why would some species have additional photographs? What are some examples of these patches used?

Lines 268-278: I am confused about the purpose of the LDA here. The authors are comparing males to cryptic-morph females? Or male-morph females to cryptic-morph females? If the former, were male-morph females also included? And why were only Class 1 females used? I thought Class 1 and 2 were lumped together in prior analyses?

Lines 281-282: What is the sample size for this analysis? Is it the larger dataset or smaller again?

Lines 292-293: Were the climate variables also included as fixed effects in this model?

Lines 321-322: It is curious that FPL evolved twice in the Bee clade, when only 1 species exhibits FPL. Any thoughts on this?

Lines 340-352: The authors mention *Calypte anna* here and *Calliphlox amethystina* in the ESM but said that only 1 bee species exhibited FLP. Is this a typo?

ESM - It would help if the authors labeled the different textual parts of the ESM as Text S1, Text S2 etc., so that when they refer to the ESM in the main text, the reader knows exactly what part is referenced.

ESM - Sensitivity analyses: The difference between an 85% and 90% chance to detect an androchrome female does not seem that big of a difference to me, especially for one to be labeled as liberal and the other conservative. Is there a reason for using these numbers? Why not 95% or 99% for the conservative? Or 70% for the liberal?

ESM - Table S1: to better correspond with the text, I would change the "yes" and "no" classifications to "presence" and "absence"

Author's Response to Decision Letter for (RSPB-2020-3004.R0)

See Appendix B.

Decision letter (RSPB-2020-3004.R1)

28-Jan-2021

Dear Ms Diamant

I am pleased to inform you that your manuscript entitled "Male-like female morphs in hummingbirds: the evolution of a widespread sex-limited plumage polymorphism" has been accepted for publication in Proceedings B.

Open Access

You are invited to opt for Open Access, making your freely available to all as soon as it is ready for publication under a CCBY licence. Our article processing charge for Open Access is £1700. Corresponding authors from member institutions (<http://royalsocietypublishing.org/site/librarians/allmembers.xhtml>) receive a 25% discount to these charges. For more information please visit <http://royalsocietypublishing.org/open-access>.

Paper charges

Sincerely,

Dr Sarah Brosnan
Editor, Proceedings B
mailto: proceedingsb@royalsociety.org

Appendix A

Dear Dr. Brosnan,

Thank you for giving us the opportunity to revise our manuscript. We appreciate the time and energy put into critiquing our work. We have responded to the constructive feedback from reviewers below and incorporated substantial re-analyses into our manuscript. Although most of our findings were not altered, we did find a slight yet significant negative association between mean precipitation and FLP using the revised method suggested by Referee 2, for which we are grateful. We have also altered the language around the strength of our adaptive explanations and our rejection of the neutral hypotheses, making the paper stronger methodologically and hopefully clearer. We have color coded our responses such that comments are in black and our responses are in **blue and bolded**. We are attaching two versions of the manuscript, one with tracked changes and a clean version with no tracked changes. Line numbers are in **red** and refer to the clean document. We look forward to your and the referees' comments and consideration. Thank you, again.

Sincerely,

Eleanor Diamant

Dear Ms Diamant:

We have now received two reviews of your manuscript RSPB-2020-1963 entitled "Male-like female morphs in hummingbirds: the evolution of a widespread sex-limited plumage polymorphism." Both reviewers, the Associate Editor, and I are very enthusiastic about many aspects of your manuscript, however the reviewers and the AE have all recommended that substantial revisions are necessary. In particular, one reviewer, the AE, and I would like to see stronger evidence in favor of the adaptive explanation if that is to be the one supported. With this in mind we would be happy to consider a resubmission, provided the comments of the referees are fully addressed. Please note that this is not a provisional acceptance.

- 1) A 'response to referees' document including details of how you have responded to the comments, and the adjustments you have made.
- 2) A clean copy of the manuscript and one with 'tracked changes' indicating your 'response to referees' comments document.
- 3) Line numbers in your main document.
- 4) Data - please see our policies on data sharing to ensure that you are

complying (<https://royalsociety.org/journals/authors/author-guidelines/#data>).

Sincerely,

Dr Sarah Brosnan
Editor, Proceedings B
mailto: proceedingsb@royalsociety.org

Associate Editor

Comments to Author:

The manuscript has been reviewed by two experts in the field. Both reviewers like the topic and fit for the Journal. While Reviewer 2 recommends that only minor revisions are necessary, Reviewer 1 finds that there are significant issues with the analyses that lead to the overstating of the conclusions regarding the adaptive value of the female-linked polymorphisms. I agree with the Reviewer that stronger evidence is needed before the adaptive interpretation would be supported. I would like to give you the opportunity to respond to these concerns. However, please note that we are hoping for a substantive effort to address these concerns, and that there is no guarantee of eventual acceptance of this manuscript. If you choose to send us a revised manuscript, it will be sent back to the original reviewers, should they be available and willing to view a revision.

In addition to the comments of both reviewers, I would add that I felt that the structure of the manuscript could be clearer. There seem to be 4 aims given in the last paragraph of the introduction, but they are buried and hard to pull out. These do map on to 4 results subsections, but the language used in the introduction and the results section to describe these is not the same. Similar structures and headings are then used in the methods and results sections, yet they do differ, which makes the manuscript unnecessarily hard to follow. For example, a Methods subheading "Morphological correlations of androchromic variation within and across species", followed by a Results subheading: "Morphological correlations of androchromic variation: a test of pleiotropy". It would be much easier to track the manuscript if exactly the same structure, subheaded by exactly the same (not just similar) words were used in each section for these 4 aims (starting in the introduction) wherever possible.

Thank you for this helpful feedback on how to improve our manuscript. We have altered the language in the final paragraph of the introduction (lines 99-123) to more directly relate to subsections in the Methods and Results, and to clarify our aims. We have also edited subheadings in the Methods section to match those in the Results section (lines 150, 214, 241, 280).

Reviewer(s)' Comments to Author:

Referee: 1

Comments to the Author(s)

Dear Authors,

I enjoyed reading your paper. It is an interesting subject for all the reasons you describe. In general I think the investigation has been well conducted and reported but there are several areas where improvements could straightforwardly be made. I will preface my detailed comments by saying I am not particularly knowledgeable about hummingbird ecology and evolution, so some of my points might reflect my own lack of understanding, in which case they could be resolved with inclusion of more natural history and background in the text.

While it is always difficult to draw conclusions from null results and the predictions (L105-116) for the non-adaptive hypotheses could be described more clearly, I think they are good tests of intralocus conflict and pleiotropy hypotheses. My only real query on this aspect was whether there is dimorphism in traits other than bill size that might be involved in pleiotropic effects on F plumage?

Thank you for your constructive feedback. To help make the biology of hummingbirds clearer, we have added some relevant information on hummingbird ecology and evolution in the Introduction (lines 110-112) and Discussion (lines 411-413). Briefly, bill size and shape are the most impactful trait when thinking about hummingbird ecological niche breadth and intra- and interspecific specialization in foraging. Although wing size (particularly wing loading) and body size are also known to be sexually dimorphic in many species of hummingbirds, these traits seem to be secondary to differing foraging strategies that result from selection on resource use and bill shape/size. As such, we would expect bill differences to be the most important morphological trait and to also occur when other sexually dimorphic morphological traits occur. Even so, we did look at wing size (originally and in this version of the manuscript), which could be a proxy for body size as well, and did not find evidence for the pleiotropy hypothesis. We have now more explicitly stated that finding in the main text (lines 355-357 and lines 419-420) because it was not clearly stated outside of the ESM in the prior draft.

5. The investigation of the adaptive hypothesis is much more exploratory. This is fine in principle if it is flagged as such throughout, but this only happens in some places and not others. In the introduction there is no explanation for why the specific socioecological variables were selected, and what relationships might be expected under adaptive hypotheses. However in the abstract and discussion weak associations between FLP and migration and social dominance are taken to support a “social benefits” explanation of FLP. This seems very preliminary. In my view, the conclusion “we were able to demonstrate that social selection maintains ... a female limited polymorphism” is not clearly supported by the presented evidence. In particular I am concerned that this evidence rests on the decision taken at the end of the methods to exclude monochromatic species. This is not explained in the text. As this is the only analysis that returns (marginally) significant results, the reader will want to know why the decision was taken to explore only this subset of data. What theoretical justification is there for removing

monochromatic species? Were any other subsets also explored? Was the decision to explore the subset taken before or after results including the monochromatic species were known? There is a fine line between exploring a dataset and dredging, so I think the MS needs to be very clear about the analysis decisions.

Thank you for pointing out areas that needed clarification. We by no means want to imply “dredging” and have provided explanations for analytical choices where you have pointed out they were lacking. Particularly, we did not explore subsets other than excluding monochromic species. We had planned to conduct the analysis including monochromatic species in the “no FLP” group and excluding them in the “no FLP” group from the outset. Our concern was that monochromic species may be the result of different selection pressures and behavioral traits amongst dichromic species, such that potential differences in comparison to FLP species would be obscured by grouping the two. Second, since we were unable to differentiate between monochromic drab and ornamented species, especially as there may be important implications on social and ecological traits, we were concerned that including monochromic species might further confound potential results. We believe that the comparison between FLP and dichromic species better assesses why a species may have ornamented and non-ornamented females compared with a species where all females are cryptic, and the species generally is cryptic. We have added clarification within the main text outlining this justification (lines 297-306). We have updated the language around the adaptive explanations to reflect the exploratory nature of the analysis and results, which should also help to avoid misinterpretation about the strength of adaptive findings (lines 119, 416-417, and 446-450).

Related to this point is the choice of behavioural and morphological traits to test the adaptive hypothesis. If the social signalling hypothesis is the prime adaptive hypothesis then why measure migratory tendency and not a variable more directly related to territoriality and social behaviour? Other traits that might be more appropriate that might be able to be analysed in a revised version of this paper include position in interspecific dominance hierarchy (or does this always depend on body size?) and the number or similarity of sympatric species. In general, I would like to see more explanation and context given to how the adaptive hypothesis is investigated, and the results of this analysis. At present, while I appreciate you are clear more work needs to be done on the social signalling hypothesis, the current explanation for why a fairly weak association with migratory tendency supports it seems very convoluted and tenuous. It is possible to think of other explanations for an association of FLP with migration (not forgetting this is only among dichromatic species), for example disruptive selection on F plumage arising from different selection pressures in summer and winter ranges.

Thank you for your thoughtful feedback on our hypotheses and inferences. We agree that our social explanation is more preliminary. We unfortunately do not have data on dominance hierarchies and positions therein across the large numbers of species necessary for a comparative analysis of this sort. Future research would gain from looking at interspecific diversity across ranges and seeing if this diversity is associated with the existence of FLP within and across ranges, which we now suggest in our discussion (lines 428-430). In our re-analysis, we have found an association with lower mean precipitation (lines 360-363), which adds some strength to the socioecological explanation. Specialization

decreases with precipitation in studied hummingbird systems, which presumably increases interspecific competition, adding preliminary support for a socioecological working hypothesis. Nonetheless, we have altered language throughout the manuscript to reflect the uncertainty and needed work around this hypothesis (lines 119, 416-417, and 446-450).

There are other places in the introduction too where I would like to see arguments better developed to help the reader follow including L59-60 – why might it provide insights?; L87-90 why does this work suggest adaptation but not rule out non-adaptive explanations?; L97 why is determining evolutionary history necessary for inferring function?

We appreciate feedback on how our arguments can be better developed and aid the reader in understanding the context of our research. We have added clarification and development for these points in our arguments (lines 61-62, 90-95, 95-97).

The other part I had difficulty following was in quantifying FLP. Being unfamiliar with Bleiweiss I found description of this confusing. The text and SI should stand alone. Specimens both differ in the number of androchromic patches and the area of each patch with androchromic colour and this determines % score – correct? Figure S1 didn't especially help – something like Fig S4 might be more useful if the same patches were used? I also wondered why convert the % to a 4-level ordinal variable. Wouldn't it be better to do uni/bimodal classification on the continuous scores? Finally, without methodological detail on how the “whole body” image analysis was conducted, its inclusion doesn't address the issue that subjective scores might quantify variation as well as a more objective analysis (the objective analysis could just be done badly). I suggest removing it. Scoring discreet patches subjectively is perfectly appropriate given this study's goals. If it stays then it must be properly reported.

Yes, your understanding of our approach is correct. We have only collected data in an ordinal classification; although a continuous percentage score would have been better to use for uni/bimodal classification. Nonetheless, we have now added some clarification for why we decided to classify ordinally (lines 158-159) and clarification on androchromic patches (lines 160-161) that we hope will aid the reader. Because sexually dimorphic plumage patches are often smaller and not generalizable across species – and are particularly difficult to differentiate when they are the same color as surrounding plumage patches – Figure S4 (now **Figure S3) is not generalizable to our qualitative classification scheme. We have also deleted previous Figures S1 and S2 for the reasons you have outlined.**

Another aspect that it might be possible to straightforwardly address is handling of phylogenetic uncertainty. In ancestral state reconstructions and identification of the number of independent origins of a trait the tree topology is very critical – if a tree block were available it would be great to run analyses across this to get bounds on estimates.

Thank you for bringing up this necessary issue of phylogenetic uncertainty, which we had failed to consider in our prior MS. We have altered some language around the independent evolutionary events of FLPs to reflect implicit uncertainty of ancestral state reconstruction (line 318-320). Although we do not have access to tree blocks, we have conducted two

analyses to delve into potential uncertainty. We ran an analysis randomly pruning the tree by 10% of the branches 1000 times and computing root state likelihoods (following your comment below by not fixing the probabilities at the root). We found only slight variation, and the standard deviation of root states was relatively low (0.497 ± 0.028 SD likelihood sexually dichromatic, 0.304 ± 0.029 likelihood SD sexually monochromatic, and 0.199 ± 0.023 SD likelihood FLP). We present these methods (lines 237-239) and findings (lines 325-328) to readers for their consideration.

In addition, we have also now performed stochastic character mapping using a Bayesian method 100x to visualize a distribution of stochastic character map histories:

Here, 100 stochastic character maps are mapped onto one stochastic character map. Red refers to FLP; black refers to monochromatism; grey refers to dichromatism. Although this does not give estimates of bounds explicitly, it qualitatively demonstrates that there is a

degree of uncertainty in the ancestral state estimation. Nonetheless, the mapping does qualitatively show that transitions between states, particularly to FLP, seem to occur independently closer to tips in most clades. We are presenting this figure here rather than in the ESM, unless the editor requests us to or the referees prefer to have it in the ESM.

I hope these comments are useful in improving the manuscript. I also make a few minor suggestions below.

Yours sincerely,

Will Allen

L48 -50 Does this sentence intentionally ignore selection consequence on male preferences, or is it just unclear? My understanding is that theory and evidence currently point to the possibility of M choice leading to the evolution of F signals in polygynous species.

We have now added information about male preferences (line 51). You make an important point that, yes, theory and evidence do point to the possibility of male choice driving the evolution of female ornaments, although this seems to not be as strong a driver in polygynous species as socioecological explanations, specifically, for traits that are not direct indicators of fertility (such as plumage).

L67 Inconsistent reference formatting.

This has been edited (line 69).

L99 I suggest writing this paragraph in the present tense.

We have updated this paragraph to present tense (lines 99-123).

Specimens: The size of the sample is very impressive. I think it would be useful to present a histogram of species' sample sizes in the SI.

This is a good suggestion. We have added a histogram of species' sample sizes in the ESM (Figure S1), highlighting female specimens as their sample sizes are most pertinent to the reliability of our findings.

L167 I think it is Hartigan's test.

We have corrected this typo in the revised MS (line 169).

L177 Novel species or increasing the sample size of species already in the dataset?

We have now clarified in the text (line 179-180) that these are increasing sample sizes of species already in the dataset.

L219 I disagree with the decision to fix the probabilities at the root. There are many evolutionary scenarios where FLP could have been present at the root. Why not just let the data decide?

Thank you for this feedback. We have conducted a re-analysis of all of our ancestral state reconstructions and transition rates. Our new analyses found similar results: the equal rates model was most supported for the tree we present in text, and no matrix produced a transition rate model in line with our null hypothesis. Furthermore, the estimates for independent evolutionary events of FLP are the same as our prior results, although in response to your comment above, we have altered our language to reflect potential effects of tree topology on that estimate. These results indicate that the sexually antagonistic intralocus conflict hypothesis is not supported by our data. We present this re-analysis of our data in the revised MS.

L232 “reconstruct trait evolution on trees”

We have fixed this wording accordingly (line 230).

L249 Report the mean sample size of the three groups not the target sample size.

We have altered this to report the mean sample sizes across sampled species (lines 249-252).

L261 It is a shame that a UV sensitive camera was not used, as hummingbirds are UV sensitive. This issue (that specimens may differ in ways invisible to the camera and experimenter, but visible to hummingbirds) should be acknowledged in the text.

Our study would have benefited from UV sensitive cameras. However, we note that we did use a UV camera in another study looking at only one hummingbird species and compared UV to non-UV images. No additional plumage patterns found in visible channels were revealed in UV channels. In addition, UV was redundant with visible blue (>97% correlation) in plumages with high UV reflectance. We also discuss further research that demonstrates correlation between visible and UV coloration (lines 263-266).

L282 aggressive towards conspecifics, heterospecifics or both?

We have now clarified that the dataset we used defined dominance as aggression towards heterospecifics (lines 285-286). Nonetheless, unpublished work by one of our authors supports that androchromic plumage in females is used in conspecific and heterospecific resource conflict (Falk et al., unpublished data).

Table S1 suggests that when a liberal 'yes' threshold is used the ARD model has a lower AICc score. This is not what is reported in the text. Is this an error or have I misunderstood?

We apologize for an important typo in the text – we reported the conservative “yes” threshold and liberal “no” threshold, which was our intention and the tree we’ve used for

all the phylogenetically-informed analyses throughout the main text, yet we made a mistake in accidentally referring to this as the “liberal presence”/”conservative absence” thresholds in the “Quantifying female-limited polymorphism in hummingbirds” results subsection. We have fixed this error in **line 311**.

L327 Is 2/16 significant associations beyond what would be expected by chance (multiple comparison problem)?

We have altered our alpha value to now take into account the multiple comparison problem (**line 276**). In reference to a comment by Referee 2, we are presenting associations between bill length and LD1 in our main text (**lines 340-352**). Two correlations are significant when taking into account Bonferroni’s correction (**lines 346-347**).

22. L357 I assume all monochromatic species all have ‘female like’ cryptic colour but this should perhaps be pointed out as if all or some have male like colour the predictions would change.

Thank you for bringing up this important point. We have added clarification into the revised manuscript (**lines 388-394**). The vast majority of monochromic hummingbird species qualitatively look more cryptic or dull, with the exception of a couple of species we refer to in our discussion (e.g., *Panterpe insignis* and *Eulampis jugularis*). As such, we think that our prediction is supported. However, we were unable to categorize species as monochromic ‘cryptic’ and monochromic ‘bright’ as has been done in other groups given that traits used to define ‘ornamented’ species in the literature (e.g., not brown, exhibited spots or stripes, exhibiting iridescent patches, etc.) also describe monochromic species in hummingbirds and ‘dull’ females in sexually dichromic hummingbird species.

Referee: 2

Comments to the Author(s)

(this markdown-formatted review is also available in compiled format as an attachment)

Review for manuscript 'Male-like female morphs in hummingbirds: the evolution of a widespread sex-limited plumage polymorphism' (for ProcB)

In this manuscript, the authors address the question of the existence of sex-limited polymorphism with male-like females (in terms of plumage colouration) in hummingbirds. This topic has been discussed multiple times at the intraspecific level, especially by Bleiweiss, but surprisingly, no study had yet approached it from the interspecific angle, by looking at the whole hummingbird family. Here, the authors use the fairly recent advances in the reconstruction of the phylogeny of the family and gather an extensive dataset from various museums to try and find out which evolutionary mechanisms drive the evolution of female-limited polymorphism.

Overall, I think the manuscript is of good quality. The science seems sound and it reads well. Importantly, it does fit quite well with ProcB format. I think some specific points could be further clarified, and maybe sometimes require additional/different analyses but I'm willing to be convinced otherwise.

Thank you for your thoughtful words.

Line by line comments

- L87-88:

> "Together, this work suggests an adaptive function of female androchromic plumage rather than a non-adaptive genetic correlation with males"

I think you're being too fast here. As you mention in the following sentences, this is compatible with the pleiotropy hypothesis, which is not adaptative. Something like "Together, this work tends to disprove the sexually antagonistic intralocus conflict hypothesis." This also has the advantage of repeating verbatim the name of one of your hypothesis, which helps the reader memorizing the various alternatives.

Thank you for this point and idea of how to help the reader remember our hypotheses while they read. We have trimmed this paragraph (lines 79-97), including the line you are referring to, and altered the language.

- L106-108:

> "(2) higher transition rates from sexual monomorphism to FLP and from FLP to sexual dimorphism than from sexual dimorphism to FLP or from FLP to sexual monomorphism."

I didn't really understand the reason for this at first and it's only later in the manuscript (L359)

that I understood. Your explanation in the discussion is better in my opinion and the mention of the "resolution of intralocus sexual conflict" is absolutely crucial to fully understand your predictions.

We appreciate the comment on how to better our writing and clarify our predictions for the reader. We have added mention of the resolution of intralocus sexual conflict and better laid out our predictions in a way that hopefully is more understandable to the reader (lines 104-108).

- L132-133:

> "including those in which sexing methods were unknown (63% of samples)"

Since this is a weak point of your study, it may be worthwhile to mention in this 'Sexing' subsection that you try to take into account this uncertainty in your analyses (by using different thresholds).

This is an excellent suggestion. Following your helpful suggestion, we have mentioned how we are taking uncertainty into account at the end of the 'Sexing' subsection (lines 146-148).

- L141:

> "those with unknown sexing methods across sexually dimorphic species for which there were ≥ 10 females with unknown sexing, ≥ 10 females with gonad data, and at least one androchrome observed (N = 41 species)."

By using the data you provided, I find N = 32 (see attached script). It may be because I misunderstood your dataset so please double check and either correct this number or make your data more re-usable (by using more machine-friendly column names and by providing a key explaining the meaning of each column).

Yes! You are right. We had an error in the text. Thank you for catching this issue; we have revised this (line 140) and double checked our numbers across the MS, particularly as we've conducted new analyses. We greatly appreciate the feedback on issues with re-usability in our dataset. We have re-named columns in our datasets so that they are machine friendly and consist of all relevant variables in this study, as well as attaching clear keys. We have updated our Dryad files accordingly.

- L149:

> "by one researcher"

Given the size of the dataset, I understand it was not possible to have multiple people to rate the specimens. But since I'm assuming the first author of this study did the observation, has you given a thought about how your preconceived explanations for FLP might have influenced your

observations?

You raise an important point. We know that there are trade-offs in these kinds of decisions. The first author personally did not expect the widespread prevalence of this trait across hummingbirds and did not have particular preconceptions. Nonetheless, there could of course be bias, particularly affecting the labeling of the extent of androchromic plumage within individuals and potential at the species level. We are hopeful that the ordinal classification scheme and our sensitivity analyses are strong enough to account for potential bias towards finding or exaggerating androchromic variation. This bias did not affect quantitative analyses testing pleiotropy and likely could not affect transition rates that tested non-adaptive antagonistic intralocus conflict.

- L151-153:

In the supplementary data that you provided, what does "1,4" mean in the `Class` column?

We have added a key. Those were monochromic individuals. We still systemically went through monochromic individuals and monochromic subspecies of otherwise dichromic species.

- L269-270:

> "We normalized bill length by dividing it by wing length to account for body size differences among species."

I don't think wing length is a good body size index in hummingbirds. Why not use the full body size? You seem to have this data since you mention it in the next subsection. I'm not sure scaling is necessary at all anyways. I'd instead present "bill length by LD1" that you included in ESM.

We appreciate this feedback and have also reflected on these choices. We agree that not scaling bill sizes is most appropriate for this test, particularly given that bill shape and size – not relative to body size – underlies ecological niche breadth in hummingbirds. We have moved the “bill length by LD1” results from the ESM to the main text and moved the scaled analyses to the ESM for completeness. We have also added more information about wing length by LD1 in the main text, as wing length and/or body length is also known to be sexually dimorphic in hummingbirds (lines 111-113, 409-413).

- SI L40-41:

> "We extracted the latitude and longitude of the centroid of the range of each species using range maps."

This doesn't seem like the most robust way to do this. For starters, how did you did with disjoint ranges? The centroid might land in an area where the species is absent. Additionally, even if the range does not have disjoint components, the centroid may be in the most extreme area in terms

of climate. A much more robust option in my opinion would be to extract climatic data from the whole range, and *then* take the mean/median.

Thank you for suggesting this important methodological improvement. We have re-analyzed our climate data using an aggregate mean of the mean temperature, mean precipitation, temperature predictability, and precipitation predictability. Indeed, we found a significant negative association with the aggregate mean precipitation in this re-analysis and have presented it in our results section (lines 360-363), as well as further details in the updated methods and results in the ESM.

Appendix B

Dear Dr. Brosnan,

We appreciate the opportunity to revise our manuscript in response to the constructive critique from a third anonymous reviewer. We have revised our manuscript in line with the overarching critiques, line-by-line critiques, and suggested re-analyses. Specifically, we have incorporated testing symmetrical transition rates and increased our sensitivity analyses into our test of non-adaptive intralocus conflict. We have also modified our language when discussing the implication of our results and are grateful for a third-person referee bringing up important caveats to our initial conclusions and areas needing clarification. We have color coded our responses such that comments are in black and our responses are in **blue and bolded**. We are attaching two versions of the manuscript, one with tracked changes and a clean version with no tracked changes. Line numbers are in **red** and refer to the clean document. We look forward to your and the referee's comments and consideration. Thank you, again.

Sincerely,

Eleanor Diamant

Associate Editor Board Member

Comments to Author:

The manuscript has now been assessed by a further reviewer. This reviewer is (like the original two), also supportive of the manuscript, and thinks that it would make a strong addition to Proceedings B. The reviewer has made some suggestions about the modeling approach, and has also made a number of suggestions for additional clarifications. I think that these could usefully be addressed in a further revision. I am optimistic that the authors will be able to address all of these comments.

Thank you for this feedback and support of our manuscript. We have expanded our modeling approach in our ancestral state reconstructions to incorporate Referee 3's recommendations and are grateful to have the opportunity to add clarification and revision where needed.

Reviewer(s)' Comments to Author:

Referee: 3

Comments to the Author(s).

In this study, the authors tested how widespread FLP was in hummingbirds and then tested what might predict the distribution and evolution of this trait. They found that FLP was widespread in hummingbirds and likely evolved independently in each major tribe. They then tested the non-adaptive hypotheses of female colouration being driven by sexual selection on males or pleiotropy. They concluded that these hypotheses were not supported. Then the tested three ecological variables to see if these traits predicted FLP evolution and found support for migration and climate to predict FLP evolution.

Overall, I found this manuscript to be interesting and on a topic of importance. I applaud the authors for the breadth of their examination and testing of many different hypotheses in one study. I think this paper has great potential and could be a strong contribution to Proceedings B. However, I have some reservations about the analyses and interpretation of the results that should be addressed. I also had several questions about the methods that should be clarified to help with the interpretation of the results.

We appreciate your kind words and for pointing out where we need revision both within the analyses and our interpretations. We are hopeful that our revision helps add clarity, accuracy, and precision throughout the manuscript.

1. First, I am not fully convinced that sexually antagonistic intralocus conflict hypothesis has been rejected by the current analyses. Specifically, I'm not sure if the results from Lines 330-338 & Lines 378-397 alone are fully convincing that the equal rates model demonstrates that FLP is not a transition state. Did the authors consider using a symmetrical rates model (SYM), as the ARD model can be heavily penalized through AIC due to the many different estimates in the model. I think the authors need to consider more evolutionary models than the equal rates and all rates different models and should also look at the number of transitions between each of the different states over time. Perhaps utilize similar analyses to Ligon et al. 2016 (Evolution, 70: 2839-2852).

We appreciate pointing out this drawback we had considered in our analysis. As such, we have incorporated a SYM model in addition to ARD and ER models (lines 200-203; ESM Text S4; Table S1; Figure S5-S14) for all of our sensitivity analyses and the analysis presented in the main text. In this re-analysis, our null model was not supported and, broadly, we found the equal rates model most supported by half of our analyses and the SYM model supported by three of eight analyses. There, the highest transition rates were to and from FLP and dichromatism. One sensitivity analysis (5% "presence" threshold; 95% "absence" threshold) led to similar support across ER, SYM, and ARD models, though the ARD model in this case was not similar to our null model and had the highest transition rates from dichromic to FLP (Table S1; Figure S14).

2. Secondly, the statements from Lines 351-352 & 406-410 seem mis-leading to me, unless I am reading the results from the ESM wrong, as it seems there were quite a few correlations between wing length/bill length with LD1. Even if these ESM results are marginally significant under the corrected p-values they must be interpreted, since the authors are also interpreting a marginally significant result with regards to FLP and social dominance (lines 418-419).

You are correct that there are some significant and marginally significant correlations with wing and bill length in the ESM and main results, though this is not a pattern across most species. We appreciate noticing potential bias in our language and interpretations as we do not want to mis-lead readers. We have now altered our discussion of these results (lines 329-332) and opened up the possibility that pleiotropy plays a role in the evolution of FLP within individual species in our discussion, though it does not seem to be an overarching pattern in the group as a whole (lines 392-395 and 400-402).

3. Thirdly, I do not think I understand how relationships with migration and precipitation indicate that social interactions and territoriality are driving the evolution of FLP (Lines 419-423). These are interesting relationships that are definitely worth exploring in the discussion, but the current discussion of these results seems lacking and shoe-horned to fit a specific social selection explanation. Also, what were the sample sizes for the ecological analyses? If they used the larger dataset, then how can the authors reconcile the differences in sample sizes between the LD1 analyses and these analyses?

We have expanded the scope of the discussion so that it is not limited to shoe-horning a social selection explanation (lines 400-416) and have also altered language in our abstract and our conclusions (lines 37-40 and 431-436). We have added further information on how migration could impact competition (lines 267-269 and 408-410). Briefly, some hummingbird species have migration separated by sex, potentially decreasing competition amongst some migrant species. Precipitation and ecology seem to influence selection pressures and competition in hummingbirds, though we agree that linking these traits in a social selection hypothesis is too speculative based on the data that we have and considering your point above on not completely ruling out pleiotropy across species. Our sample size for the ecological analysis was the whole dataset, which we now are reporting in the (lines 255 and 263). We cannot completely reconcile the sample size differences, which is why we have now altered our language and interpretations in our discussion to further open the possibility that adaptive or non-adaptive pleiotropy might play a role in the evolution of FLP in at least some hummingbird species (lines 392-395 and 400-402), adding needed nuance to our discussion. Though, we think that socioecological hypotheses should be further explored in future work (lines 413-416 and 434-436).

I had the following addition comments as well:

4. Lines 114-115: Would you not also expect this relationship in males? Because if there was no relationship between male colour and bill morphology, but there was a relationship between female androchromatic colour and female andromorphic bill morphology, that would not necessarily support pleiotropy, in my mind.

Yes, we should expect this relationship in males as well as females if pleiotropy exists. In the paper, we are focusing on FLP and – given the need for individuals across a continuum of coloration – we focused on female variation in our analyses and methods.

5. Lines 150-152: Perhaps include in parentheses the initials of which author or researcher scaled the plumage?

Thank you for this suggestion! We have added the researcher's initials (line 141).

6. Lines 152-155: was this classification based on differences in hue and patch size alone? Or were differences in brightness, chroma, and angle-dependence also considered? More details on what plumage features were used to quantify differences between males and females would be helpful.

This classification was based on difference in hue and patch size alone. We did not consider differences in brightness as that seemed challenging to decipher by eye across specimens for our qualitative analysis. We have added this detail to our methods section (line 143). For our quantitative analysis in a subset of species, we included angle-dependence in our photography methods and are hopeful that differences in chroma are captured by our photography methods, which has been moved to the ESM due to article length limitations (ESM Text S5).

7. Lines 163: does this mean that degree of androchromatic plumage was only classified based on their iridescent plumage? Or was the whole body used and this reference to iridescent plumage is only about determining which subspecies to use?

Androchromic plumage was determined by what plumage was different between males and females and distinguishable to the naked eye – including coloration that was not necessarily iridescent (e.g., hue and patch size as described above). We have now added clarification in the main text (lines 153-156).

8. Lines 193-203: It would be help to mention that the authors also developed a conservative absence and liberal presence thresholds and that the descriptions for each are in the ESM. I was a bit confused in reading this section the first time through.

Thank you for pointing out an area of our manuscript that needed clarification. We have changed our language here to directly state the thresholds used in the analysis presented in the main text (lines 168-172) and added a brief description of what is presented in the ESM (lines 175-177). In the ESM, we have also altered language to more directly state how we categorized species across matrices (ESM Text S4).

9. Lines 263-266: I think the authors should also acknowledge that the lack of UV reflectance measured is a limitation of their methods, instead of just writing it off as a non-issue. Especially since citation 53 is unpublished as well. Also see Burns & Shultz 2012 (The Auk, 2: 211-221)

We have now explicitly stated that this is a potential limitation and cited Burns & Shultz (2012) (lines 233-234).

10. Lines 258-260: Why would some species have additional photographs? What are some examples of these patches used?

We have added clarification on what some potential patches may be (ESM Text S5). These are typically to capture angle-dependent iridescent patches (e.g., iridescent rump or angle-dependent iridescence across the crown or chin).

11. Lines 268-278: I am confused about the purpose of the LDA here. The authors are comparing males to cryptic-morph females? Or male-morph females to cryptic-morph females? If the former, were male-morph females also included? And why were only Class 1 females used? I thought Class 1 and 2 were lumped together in prior analyses?

This section has now been revised in the main text and expanded upon in the ESM to add needed clarification (lines 240-245; ESM Text S5). We are comparing variation from cryptic-morph females to androchromic females. We are only using Class 1 females and Class 4 males to train the LDA in order to differentiate plumage variation based on the differences between our completely androchromic male individuals and heterochromic female individuals. As such, we can better quantify continuous variation when applying the LD analysis on all of the females by training the model on holotypic cryptic females and androchromic males. In this analysis we are testing pleiotropy and thus looking across female variation. If pleiotropy is supported, we would expect the more androchromic an individual is, even if slightly so, to also result in more andromorphic morphology.

12. Lines 281-282: What is the sample size for this analysis? Is it the larger dataset or smaller again?

The social and ecological analyses used the full dataset. We have now clarified this in the text (lines 255 and 263).

13. Lines 292-293: Were the climate variables also included as fixed effects in this model?

We have now edited the manuscript to clarify that we tested climate variables in a separate model with a larger sample size (line 260 and 263).

14. Lines 321-322: It is curious that FPL evolved twice in the Bee clade, when only 1 species exhibits FPL. Any thoughts on this?

We apologize for this typo. Indeed, FLP only evolved once in the Bee clade (Table 1).

15. Lines 340-352: The authors mention *Calypte anna* here and *Calliphlox amethystina* in the ESM but said that only 1 bee species exhibited FLP. Is this a typo?

Only one bee species exhibited FLP with the thresholds presented in the main text. However, other species – like *Calypte anna* – exhibited variation that met thresholds or were unable to be classified, which we included in our LD analysis. In our table headers we clarify how each species was categorized (Table 2, Table S2). If pleiotropy were to explain the existence of FLP and broadly the pervasiveness of androchromic variation, we would expect associations between androchromic variation and andromorphic morphology in species that may or may not have been categorized as exhibiting FLP in all of our analyses.

ESM - It would help if the authors labeled the different textual parts of the ESM as Text S1, Text S2 etc., so that when they refer to the ESM in the main text, the reader knows exactly what part is referenced.

Thank you for this suggestion. We have now labeled and edited references to our ESM text throughout the manuscript.

ESM – Sensitivity analyses: The difference between an 85% and 90% chance to detect an

androchrome female does not seem that big of a difference to me, especially for one to be labeled as liberal and the other conservative. Is there a reason for using these numbers? Why not 95% or 99% for the conservative? Or 70% for the liberal?

We understand the concern about the slight difference in our sensitivity analyses and agree that adding to them would benefit our interpretations and results. We went with these two cut-offs because we were concerned with potential biases of over-pruning our tree (with too conservative thresholds) or vastly increasing the risk of false negatives (with too liberal thresholds). However, to add to the sensitivity analyses, we have added 95% and 80% thresholds while also providing details on the number of tips in the final tree (ESM Text S4; Table S1; Figure S5-S14) so that readers can determine what they consider more reliable. Nonetheless, our results did not strongly differ when adding more sensitivity analyses.

ESM – Table S1: to better correspond with the text, I would change the “yes” and “no” classifications to “presence” and “absence”

Thank you for this suggestion. We have edited the table (Table S1) and hope it better corresponds with the text.